## RESEARCH ARTICLE

# Roles of protein–protein interactions and monolayer mechanics in tricellulin localization to tricellular tight junctions

Toiba Mushtaq[1,2], Jaakko Lehtimäki[2], Konstantin Kogan[1,2], Johan Peränen[1,2], Xionan Liu[2], Markku Varjosalo[2], Aki Manninen[3] and Pekka Lappalainen[1,2,*]

## ABSTRACT

Tricellular tight junctions (tTJs) seal the space between three or more cells in epithelial monolayers. These specialized tight junctions have distinct protein components, including a transmembrane protein tricellulin. However, the mechanisms by which tricellulin localizes specifically to tTJs are incompletely understood. We demonstrate that tricellulin undergoes rapid lateral diffusion along bicellular junctions but is a very stable component of tTJs. BioID proteomics identified several proximity partners of tricellulin, and knockout studies on angulin-1/LSR, occludin and afadin provided evidence that these proteins control tricellulin accumulation to tTJs to different extents and mechanisms. Tricellulin localization was disrupted in afadin and angulin-1/LSR knockout cells, although these proteins did not display similar accumulation to tTJs, suggesting that they contribute to tricellulin localization through indirect or context-dependent mechanisms. Importantly, experiments on mixed cultures revealed that defects of tricellulin localization in occludin knockout cells were affected by the proximity of wild-type cells, and treatment of monolayers with myosin-II inhibitor resulted in displacement of tricellulin from tTJs. These results suggest that, in addition to protein–protein interactions, proper epithelial monolayer mechanics are essential for stabilizing tricellulin at tTJs.

KEY WORDS: Tricellulin, Epithelium, Tight junction, Mechanics

## INTRODUCTION

Epithelial tissues form barriers, which have a critical role in maintaining tissue homeostasis in animals. These barriers control the selective permeability of ions, solutes, and macromolecules across epithelial layers, thus ensuring proper physiological function and homeostasis of the tissue (Wibbe and Ebnet, 2023). At the core of epithelial barriers are tight junctions (TJs), which serve as the primary structures to control paracellular transport by forming a continuous barrier between adjacent cells. TJs not only physically seal cells together but also facilitate cellular communication and

signaling processes. The dynamic nature of TJs is crucial to their function, allowing them to adapt to physiological changes and mechanical stresses (Cho et al., 2022; Higashi and Miller, 2017). TJs can be further divided to bicellular tight junctions (bTJs) and tricellular tight junctions (tTJs). While bTJs are well characterized for their role in sealing the space between two neighboring cells, the tTJs form at the positions in epithelial monolayers where three or more cells meet each other. The bTJs and tTJs are distinct from one another in their molecular composition. Central components of bTJs include occludin and claudin, which are transmembrane proteins that link adjacent cells to each other, and a cytoplasmic adaptor protein ZO-1, which links claudins and occludin to the actin cytoskeleton (Citi et al., 2024). On the other hand, the major components of tTJs include a single-pass transmembrane protein angulin and another transmembrane protein named tricellulin. Angulin family in vertebrates consists of three homologous proteins, from which the role of angulin-1 (also known as lipolysis-stimulated lipoprotein receptor, LSR) is the best characterized for its functions in tTJs of epithelial monolayers. Loss of angulin-1/LSR results in reduced transepithelial resistance and increased macromolecule permeability, demonstrating that angulin-1/LSR is important for maintenance of epithelial barrier (Higashi and Miller, 2017; Masuda et al., 2011). Tricellulin (also known as Marveld-2) is a four-pass transmembrane protein, which is structurally homologous to occludin (Higashi and Miller, 2017). Similarly to angulins, tricellulin accumulates strongly to tTJs and only weakly to bTJs (Ikenouchi et al., 2005).

Depletion of tricellulin in epithelial Eph4 and MDCK II cells results in defective barrier function and disorganized tricellular junctions (Ikenouchi et al., 2005; Sugawara et al., 2021; Van Itallie et al., 2010). Tricellulin knockout mice exhibit defects in the architecture and function of tTJs, which lead to compromised epithelial barrier integrity in the organ of corti. This results in increased paracellular permeability, and ultimately damage to hair cells and early onset, rapidly progressive hearing loss (Kamitani et al., 2015). In line with the knockout studies, mutations in tricellulin cause heritable non-syndromic deafness called DFNB49, and knock-in mice carrying the same mutations in tricellulin developed rapidly progressing hearing loss and disruption of the strands of intramembrane particles connecting bicellular and tricellular junctions in the inner ear epithelia (Riazuddin et al., 2006; Nayak et al., 2013). Interestingly, tricellulin and occludin may be functionally partially redundant with each other, because tricellulin-occludin double knockout cells displayed more pronounced defects in TJ architecture and permeability for ions and small macromolecules than either single knockout cell line (Saito et al., 2021). Tricellulin was also reported to interact with a Cdc42 guanidine exchange factor TUBA, and thus regulate junctional tension through Cdc42 (Oda et al., 2014). Finally, a recent study provided evidence that tricellulin interacts with

[1]Molecular and Integrative Biosciences Research Programme, Faculty of Biological and Environmental Sciences, University of Helsinki 00014, Finland. [2]HiLIFE Institute of Biotechnology, University of Helsinki 00014, Finland. [3]Disease Networks Research Unit, Faculty of Biochemistry and Molecular Medicine & Biocenter Oulu, University of Oulu, Oulu 90014, Finland.

*Author for correspondence (pekka.lappalainen@helsinki.fi)

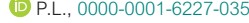 P.L., 0000-0001-6227-0354

α-catenin and hence regulates the organization of actomyosin structures at tTJs (Cho et al., 2022).

While the role of tricellulin in maintenance of tTJs is beginning to be relatively well established, the precise mechanism by which tricellulin specifically accumulates to tTJs is still incompletely understood. Depletion of angulin-1/LSR was shown to disrupt the enrichment of tricellulin to tTJs, whereas the loss of tricellulin did not affect the subcellular localization of angulin-1/LSR. Thus, angulin-1/LSR appears to be an upstream regulator of tricellulin localization to tTJs (Cho et al., 2022; Masuda et al., 2011; Sugawara et al., 2021). Moreover, depletion of occludin was shown to reduce accumulation of tricellulin to tTJs, and it was proposed that the presence of occludin at bTJs excludes tricellulin from these junctions and hence results in its accumulation to tTJs (Ikenouchi et al., 2008; Kitajiri et al., 2014). However, the precise mechanisms by which angulin-1/LSR and occludin contribute to subcellular localization of tricellulin in epithelial cells, as well as the contribution of possible other proteins and epithelial monolayer mechanics in this process, have remained elusive. Thus, understanding the dynamic nature of tricellulin within tTJs requires studying its interactions with other tight junction proteins and investigating the role of cytoskeletal forces in maintaining junctional stability.

Here, we show that tricellulin is a very stable component of tTJs, whereas it undergoes rapid lateral diffusion in bTJs. By combining proteomics, CRIPSR-Cas9 knockout techniques, immunofluorescence microscopy, and pharmacological interventions, we provide evidence that, in addition to the presence of angulin-1/LSR, proper epithelial monolayer mechanics is important for the subcellular localization of tricellulin to tTJs.

## RESULTS

### Tricellulin is a stable component of the tTJs

To elucidate the mechanism by which tricellulin accumulates to tTJs, it is important to elucidate its dynamics in TJs. For this purpose, we generated a stable MDCK cell line expressing EGFP fused to the N-terminus of tricellulin (EGFP-tricellulin) (Fig. S1A,B). Similarly to endogenous tricellulin, the EGFP-tricellulin localized to TJs in these cells and accumulated more strongly to tTJs as compared to bicellular tight junctions (bTJs) (Fig. 1A). Fluorescence-recovery-after-photobleaching (FRAP) was then applied to examine the dynamics of EGFP-tricellulin in both tricellular junctions and bicellular junctions. We first carried out a simultaneous photobleaching of EGFP-tricellulin in a tTJ and a segment of bTJ from the same field of view. Interestingly, this FRAP experiment indicated that tricellulin displays relatively rapid dynamics (lateral diffusion) in bicellular junctions, but it is nearly immobile in tricellular junctions (Fig. 1B,C). An analysis of FRAP results from a larger number of photobleaching experiments revealed that the recovery half-time of EGFP-tricellulin at bTJs is approximately 200 s. In contrast, there was hardly any detectable fluorescence recovery of EGFP-tricellulin at tTJs even after 1200 s post bleaching (Fig. 1D,E). Together, these experiments provide evidence that tricellulin is a very stable component of tTJs, whereas it displays rapid lateral diffusion at bTJs.

### Identification of proximity partners of tricellulin

Apart from interactions with angulin-1/LSR, TUBA and α-catenin (Cho et al., 2022; Higashi et al., 2013; Masuda et al., 2011; Nayak et al., 2013; Oda et al., 2014), the protein network associating with tricellulin, and restricting its localization to tTJs, is still incompletely understood. To identify protein–protein interactions of tricellulin, we

carried out a proximity-dependent biotin identification (BioID2) analysis (Roux et al., 2012) in MDCK epithelial cells using a biotin ligase fused to the N-terminus of tricellulin (Fig. 2C,D). In addition to TUBA, which is a known interaction partner of tricellulin, the BioID analysis identified several other proteins being in the proximity of tricellulin. These include a tight junction protein occludin, several actin cytoskeleton-associated proteins such as utrophin, ROCK1, ROCK2, Coronin-1C, BAIAP2/IRSp53, EPB41L1, and WASH complex subunit 3. Moreover, hits from the tricellulin BioID screen included signaling and cell polarity proteins, such as PARD3, DVL1, YES1, MID1IP1, and PKP3. Finally, afadin (AFDN), SCRIB, and MAGI3, which are known components of epithelial cell–cell junctions, were identified as high-confidence interactors of tricellulin in the screen (Boëda et al., 2023; Choi et al., 2016; Wu et al., 2000) (Fig. 2E, Table 1). From these, afadin was also earlier linked to the maintenance of tricellular junctions in *Drosophila* embryos (Yu and Zallen, 2020).

Interestingly, angulin-1/LSR, which was reported to control tricellulin localization to tTJs (Cho et al., 2022; Masuda et al., 2011; Sugawara et al., 2021) was not flagged as a significant interactor of tricellulin in our BioID-screen. However, because a previous study provided evidence that the C-terminal cytoplasmic domain of tricellulin interacts either directly or indirectly with the cytoplasmic region of angulin-1/LSR (Masuda et al., 2011) and, in our experiments, the BirA was tagged to the N-terminus of tricellulin, it is possible that the BirA was unable to biotinylate angulin-1/LSR for structural reasons. We therefore also carried out a BioID screen in MDCK cells also with a C-terminally tagged angulin-1/LSR (Fig. 2A,B). Although this screen identified several cytoskeletal proteins, including tropomyosin-1 (TPM1), myosin 5B, myosin 6A, coronin-1B, coronin-6, and WDR1/Aip1, as well as some common interactors with tricellulin (e.g. coronin-1C and several proteins involved in membrane and organelle dynamics), tricellulin was not identified as a significant interactor of angulin-1/LSR in this screen (Fig. 2E).

To examine the association of tricellulin and selected proteins identified from the BioID screen at tTJ (bold in Table 1), we applied Airyscan confocal microscopy and specific antibodies to reveal the localizations of these proteins (in xyz planes) in MDCK epithelial cell monolayers. In these experiments, we decided to focus on three proteins from the BioID screen: tricellulin (which was used as a bait), occludin, and afadin. Moreover, we included angulin-1/LSR in these cell biological studies, because angulin-1/LSR was in earlier studies reported as an interaction partner of tricellulin. For these experiments, MDCK cells were grown on 10 µg/ml fibronectin-coated high-precision glass coverslips, and the proteins of interest were studied in confluent MCDK cell monolayers. As reported before, tricellulin and occludin localized to TJs, with tricellulin being much more concentrated to tTJs vs bTJs as compared to occludin (Fig. 3A). Also, angulin-1/LSR localized to cell–cell junctions. However, based on our antibody stainings, angulin-1/LSR displayed relatively uniform localization along the cell–cell junctions, being only modestly enriched at tTJs (Fig. 3B). Thus, in these conditions angulin1-/LSR localizes to both bTJs and tTJs. This contrasts earlier studies on MDCK cells cultured on Transwell filters, where angulin-1/LSR was shown to accumulate much more strongly to tTJs (Sugawara et al., 2021). Finally, also afadin localized quite uniformly to the cell–cell junctions, without notable accumulation to tricellular junctions (Fig. 3C).

Merged images, as well as orthogonal projections from the Airyscan co-localization studies, highlighted distinct patterns of these proteins also in the Z-plane of tricellular junctions. Tricellulin

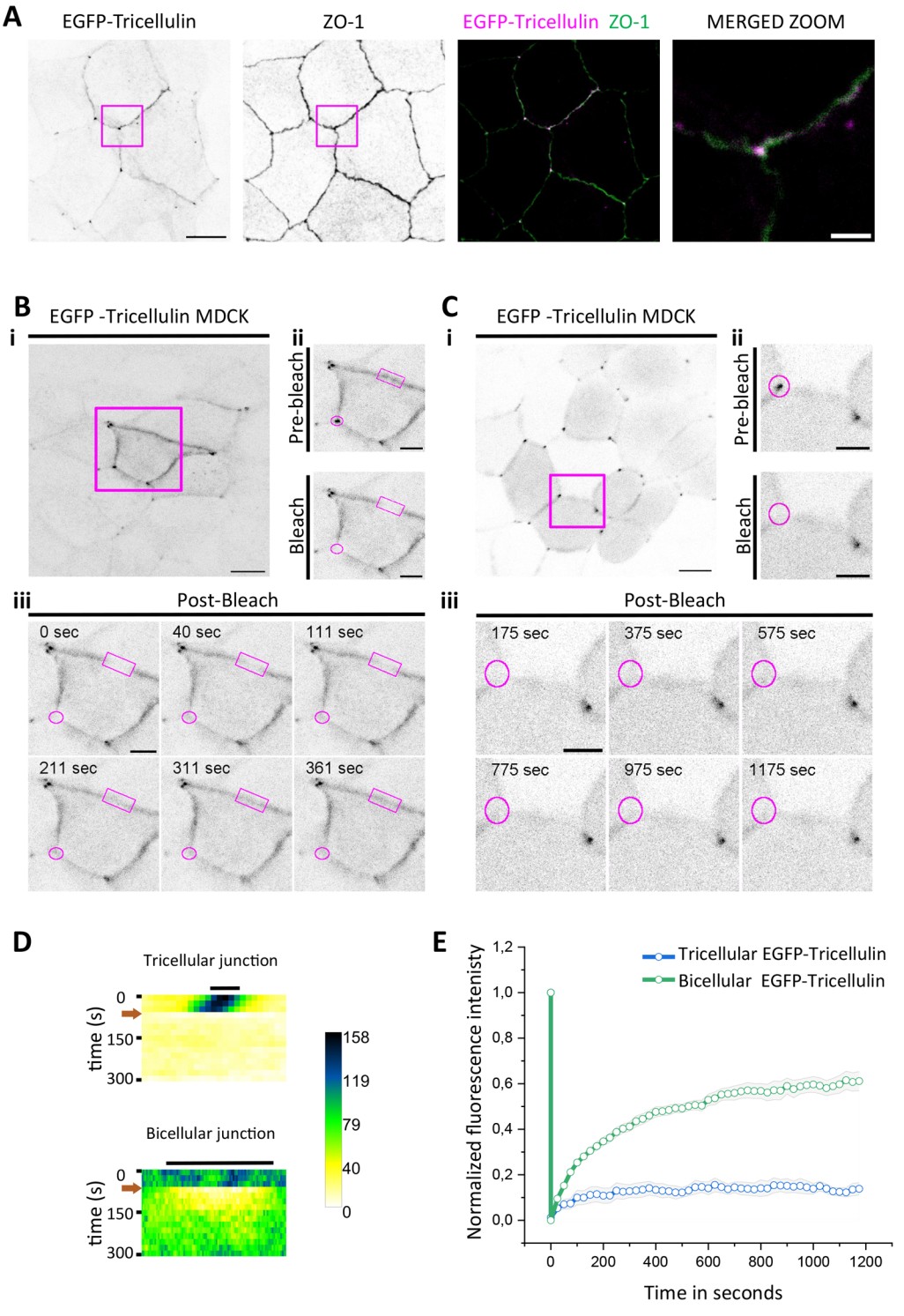

**Fig. 1. Dynamics of tricellulin in bicellular and tricellular tight junctions.** (A) Maximum intensity projections (MIPs) of wild-type MDCK cell monolayers expressing EGFP-tricellulin. Tight junctions were visualized by anti-ZO-1 antibodies. Merged images on right (the merged zoomed image is from the region highlighted by a magenta box in panels on left) show the co-localization of EGFP-tricellulin (magenta) and ZO-1 (green) at tricellular junctions. Scale bars: 10 µm and 2 µm in normal and zoomed images, respectively. (B) The dynamics of tricellulin in a stable MDCK cell-line expressing EGFP-tricellulin in confluent monolayers was studied by FRAP through simultaneous photobleaching of a region of bTJ and tTJ from the same cell. Scale bars: 10 µm. (C) Photobleaching of EGFP-tricellulin at an individual tTJ. Scale bars: 5 µm. High magnification images of tight junction segments (B-C, i-iii) in the middle panels display a photobleached cell before and after bleaching at the indicated region (magenta boxes). Smaller magenta circles and rectangles represent the photobleached regions. (D) Kymographs demonstrating the recovery of EGFP-tricellulin at tTJ (top) and bTJ (bottom) from individual FRAP experiments. (E) Recovery curves demonstrating that tricellulin is a stable component of tTJ, and dynamic at bTJ. The data are mean±standard error of the mean (s.e.m.) of FRAP experiments at tTJ and at bTJ (n=8).

and occludin displayed similar precise localization at the topmost tight junction plane (Fig. 3A), whereas the localizations of angulin-1 and afadin also extended towards the basolateral region of the tricellular junction (Fig. 3B,C).

In conclusion, the BioID-proteomics and immunofluorescence microscopy studies demonstrate that tricellulin is in close proximity to occludin and afadin at cell–cell junctions, but neither one of these other proteins displays similar prominent accumulation to tTJs as tricellulin. Moreover, although both tricellulin and angulin-1/LSR accumulate to cell–cell junctions, their localization patterns are quite different from each other in MDCK cell monolayers cultured on coverslips, and our BioID-screens did not provide evidence of these proteins interacting with each other.

### Angulin-1/LSR, occludin and afadin knockouts affect the subcellular localization of tricellulin

Previous studies on cell-lines and transgenic mice provided evidence that both angulin-1/LSR and occludin are important for the proper localization of tricellulin to tTJs (Higashi et al., 2013; Kitajiri et al., 2014; Masuda et al., 2011; Nayak et al., 2013; Saito

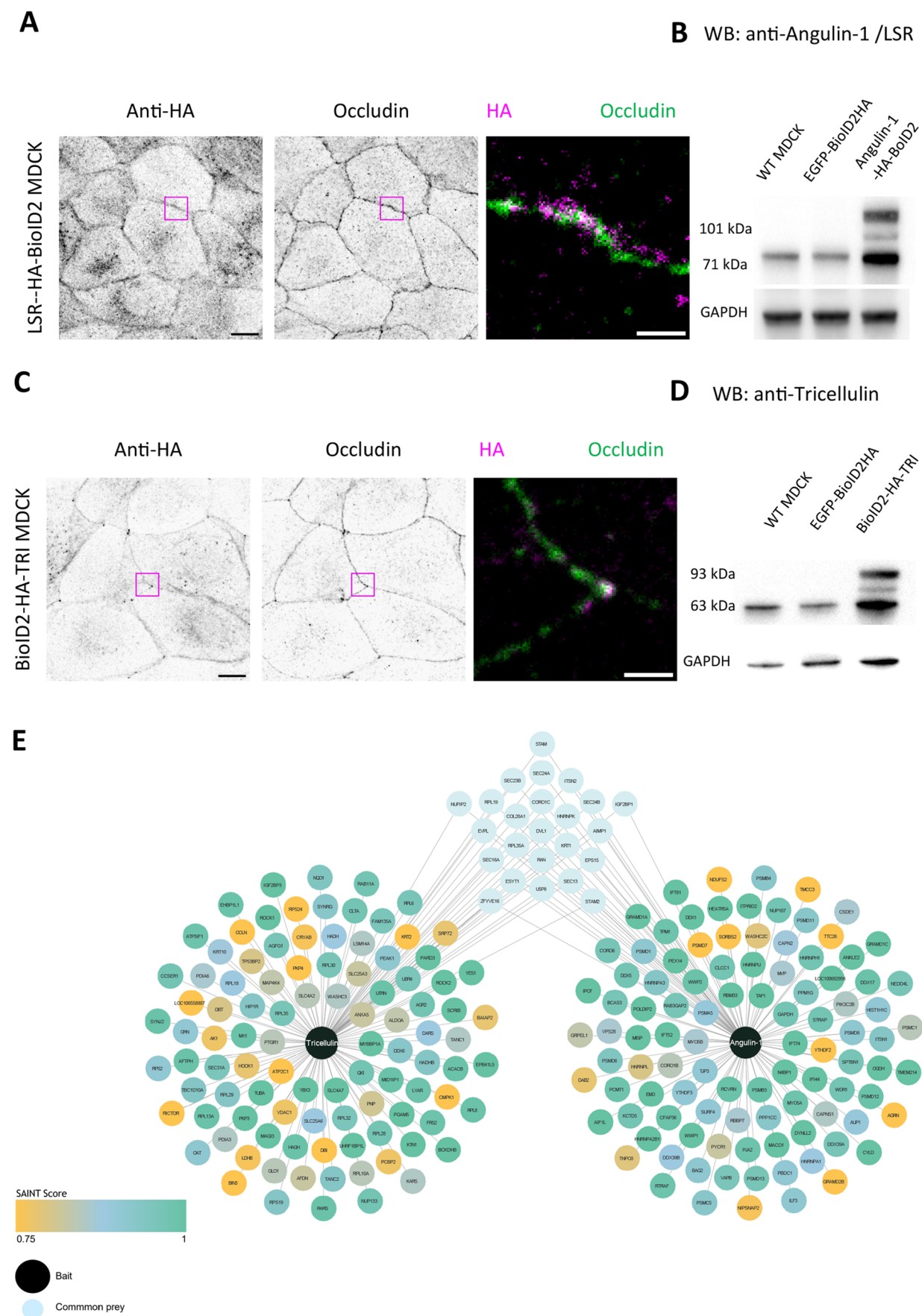

**Fig. 2.** See next page for legend.

**Fig. 2. BioID screen for interactome of tricellulin and angulin-1/LSR in cultured MDCK cells.** (A) Confocal microscopy images of MDCK cells expressing angulin-1/LSR-HA-BioID2 construct. BirA-fusion constructs were detected by anti-HA (left panel) and anti-occludin (middle panel) antibodies. Scale bar: 10 µm. Merged image of the boxed region displays the localization of angulin-1/LSR- HA-BioID (magenta) in respect to cell–cell junctions (green). Scale bars: 2 µm. (B) Western blot analysis of the levels of endogenous angulin-1/LSR and angulin-1/LSR-HA-BioID2 fusion proteins detected by antI-angulin-1 antibody in wild-type MDCK cells, as well as in cells transfected with a control EGFP-BioID2-HA-CAAX construct and angulin-1/LSR- HA-BioID2 construct (top panel). GAPDH was probed as loading control (bottom panel). (C) Confocal microscopy images of cells expressing BioID2-HA-tricellulin construct stained with anti-HA (left panel) and anti-β-occludin (middle panel) antibodies. Scale bar: 10 µm. Merged image of the boxed region (right panel) displays the localization of BioID2-HA-tricellulin (magenta) in respect to cell–cell junctions (green). Scale bar: 2 µm. (D) Western blot analysis of the levels of endogenous tricellulin and LSR-BioID2-HA-tricellulin fusion proteins detected by anti-tricellulin antibody in wild-type MDCK cells, as well as in cells transfected with a control EGFP-BioID2-HA-CAAX construct and BioID2-HA-tricellulin construct (top two rows). GAPDH was used as loading control (bottom row). (E) SAINT-filtered BioID interactome for tricellulin and angulin-1/LSR. Interaction partners within a ≥0.75 threshold of SAINT score are included in the interactome plot. The proteins are color-coded based on their respective average spectral counts with respect to the control experiment carried out with membrane bound-EGFP. Common interactors of tricellulin and angulin-1/LSR are plotted in the middle. For angulin-1/LSR one biological replicate, for tricellulin two biological replicates, and for eGFP four biological replicates were collected, and each were further split into two technical replicates prior mass spectrometry analysis.

et al., 2021), whereas the possible effects of afadin on tricellulin localization have not been reported in any experimental systems before. To study the roles of these proteins in the localization of tricellulin, as well as on the architecture of cell–cell contacts in MDCK cells, we generated tricellulin, angulin-1/LSR, occludin and afadin MDCK knockout cells by CRISPR-Cas9 approach. The success of the knockouts was confirmed by western blotting with specific antibodies (Fig. S1C), as well as by Sanger and Next-generation sequencing (Figs S2 and S3). In each case, we obtained

**Table 1. SAINT-filtered BioID interactome for tricellulin**

| Protein name | Gene | SAINT score |
|---|---|---|
| **Afadin** | **AFDN** | **0.82** |
| **Tricellulin** | **MARVELD2** | **1** |
| **Lipolysis-stimulated lipoprotein receptor** | **LSR/Angulin-1** | **0** |
| **Occludin** | **OCLN** | **0.75** |
| BAI1 associated protein 2 | BAIAP2/IRSp53 | 0.78 |
| Coronin | CORO1C | 0.79 |
| Dynamin-binding protein | DNMBP/TUBA | 1 |
| Erythrocyte membrane protein | EPB41L5 | 1 |
| Membrane-associated guanylate kinase | MAGI3 | 1 |
| Mid1-interacting protein 1 | MID1IP1 | 0.99 |
| Partitioning defective 3 homolog | PARD3 | 1 |
| Plastidial pyruvate kinase 3 | PKP3 | 1 |
| Protein scribble homolog | SCRIB | 1 |
| Rho-associated protein kinase 1 | ROCK1 | 1 |
| Rho-associated protein kinase 2 | ROCK2 | 1 |
| Segment polarity protein dishevelled homolog | DVL1 | 1 |
| Tyrosine-protein kinase Yes | YES1 | 1 |
| Utrophin | UTRN | 0.99 |
| WASH complex subunit 3 | WASHC3 | 0.85 |

The SAINT score values of the interactors with respect to membrane bound-eGFP (control) are indicated in the right-hand column. Bold indicates the selected candidates, namely afadin (AFDN), Lipolysis-stimulated lipoprotein receptor (angulin-1/LSR), occludin (OCLN) and tricellulin (MARVELD2) that we included in further analysis.

two independent knockout clones, which both displayed similar phenotypes in respect to subcellular localization of tricellulin.

As reported earlier (Higashi et al., 2013; Masuda et al., 2011; Nayak et al., 2013), depletion of angulin-1/LSR disrupted tricellulin localization to tTJs also in MDCK monolayers cultured on high precision glass coverslips (Fig. 4A; Fig. S4A,B). This is because in the absence of angulin-1/LSR, tricellulin was nearly completely displaced from tTJs and instead exhibited rather uniform localization along the tight junctions. In agreement with earlier studies, depletion of tricellulin did not affect the subcellular localization of angulin-1/LSR (Fig. S4C,D). Also, depletion of occludin resulted in diminished localization of tricellulin to tTJs (Fig. 4A). However, compared to the angulin-1/LSR knockout cells, the displacement of tricellulin from tTJs to bTJs in the occludin knockout cells was less prominent (Fig. 4D).

Interestingly, also the loss of afadin had a marked impact on tricellulin localization. Afadin is an actin-binding protein, which plays a vital role in linking the actin cytoskeleton to cell–cell junctions (Choi et al., 2016; Sakakibara et al., 2020). In the afadin knockout cells, tricellulin was displaced from tTJs in an irregular pattern, with some protein being mis-localized to bTJs, whereas other areas of the monolayer showing nearly complete loss of tricellulin from the cell–cell junctions (Fig. 4A,B). Moreover, the loss of afadin resulted in a disorganized actin cytoskeleton, which may indirectly impact tricellulin's localization to tTJs. Together, these knockout experiments demonstrate that afadin, angulin-1/LSR, and occludin all contribute, to different extents and perhaps through different mechanisms, to tricellulin's proper localization to tTJs.

### The role of epithelial monolayer mechanics to subcellular localization of tricellulin

To further examine the localization of tricellulin in the three-knockout cell-lines, we co-cultured the knockout cells with wild-type MDCK cells expressing GFP. The expressed GFP did not affect the subcellular localization of tricellulin in wild-type cells, because tricellulin accumulated to tTJs equally well in cells with and without GFP (Fig. 5A). In these experiments, the monolayer was composed of 'islands' of green GFP-expressing wild-type cells and 'non-fluorescent' wild-type or knockout MDCK cells. Experiments with mixtures of GFP-expressing wild-type cells and angulin-1/LSR, occludin, and afadin knockout cells (Fig. 5B-D) confirmed the defects in the tricellulin localization observed above from the pure knockout cultures (Fig. 4). Importantly, a closer observation of the mixed cultures indicated that the defects in tricellulin localization in occludin knockout cells were affected by the proximity of wild-type cells. This is because the knockout cells further away from the GFP-wild-type–knockout-cell border displayed a more pronounced mis-localization of tricellulin to bTJs than those occludin knockout cells that were located close to the border of wild-type cells (Fig. 5C). This positive effect of wild-type cells on the localization of tricellulin in occludin knockout cells was further confirmed by quantifying the localization of tricellulin in tTJs versus bTJs in the first line of cell–cell contacts between knockout cells (+1) (orange line in Fig. 5C) and in cell–cell junctions further away from the border between wild-type and knockout cells (≥2) (Fig. 5E). Similarly, the proximity of wild-type cells appeared to affect the tricellulin localization in afadin-depleted cells, although to lesser, and not statistically significant, extent (Fig. 5D,E).

The proximity effects of wild-type cells on tricellulin localization in the occludin and afadin knockout cells suggest that also the mechanics of the epithelial monolayer may contribute to proper

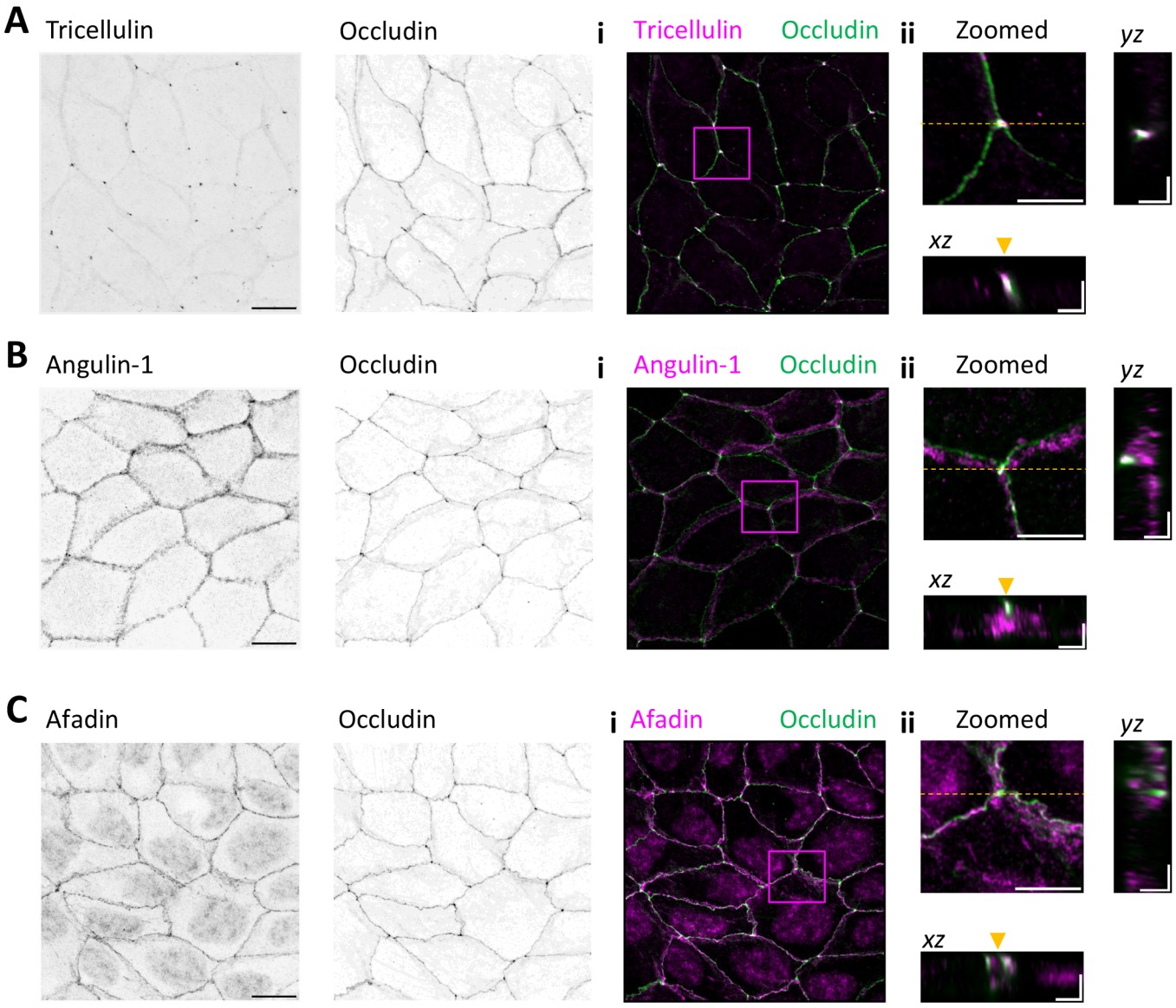

**Fig. 3. Localization of tricellulin, occludin, angulin-1/LSR, and afadin in MDCK cell monolayers on fibronectin-coated high-precision glass coverslips.** Maximum intensity projections of endogenous tricellulin (A), angulin-1/LSR (B), and afadin (C) in respect to occludin in confluent wild-type MDCK cell monolayers, as detected by anti-tricellulin, anti-angulin-1/LSR, anti-afadin and anti-occludin antibodies. Single channels are shown as inverted lookup tables (LUTs; greyscale). Merged images on the right display co-localization of occludin (green) with Tricellulin (Ai-ii, magenta), angulin-1 (Bi-ii, magenta), afadin (Ci-ii, magenta). Magnified insets (A-C,ii), along with the orthogonal x-z projections, highlight the localization of these proteins at tricellular junctions. Scale bars: 10 µm (A-Ci); 5 µm (A-Cii).

accumulation of tricellulin to tTJs. To directly evaluate this, we used a myosin II inhibitor blebbistatin to disrupt contractility of epithelial actomyosin bundles, which are responsible for contractile forces in epithelial monolayers (Smutny et al., 2010). Treatment of MDCK monolayers with 25 µM or 50 µM blebbistatin for 4.5 h led to mislocalization of endogenous tricellulin from tTJs to bTJ. In most cell–cell junctions of blebbistatin-treated monolayers, tricellulin did not display uniform localization along bTJs, but rather become concentrated in small foci that localized along the bTJs (Fig. 6A). Concomitantly, the intensity line profile analysis showed sharp peaks of tricellulin exclusively at tricellular junctions in DMSO-treated control cells, whereas in the blebbistatin-treated monolayers intensity profiles displayed multiple peaks also along the bicellular junctions (Fig. 6B). Interestingly, experiments on EGFP-tricellulin expressing MDCK cell monolayers showed similar appearance

of tricellulin foci in bTJs upon blebbistatin treatment, but angulin-1/LSR was not found enriched in these blebbistatin-induced tricellulin foci (Fig. 6C).

To further study the nature of aberrant tricellulin foci induced by blebbistatin treatment, we performed FRAP experiments on EGFP-tricellulin expressing cells after blebbistatin treatment. These experiments revealed that tricellulin displays much more rapid lateral diffusion in the aberrant foci at bTJs than the dynamics of tricellulin in the tTJs of control cells treated with DMSO only (Fig. 7). Together, the results from mixed cultures between wild-type and occludin/afadin knockout cells (Fig. 5), and studies on blebbistatin-treated monolayers (Figs 6 and 7) suggest that, in addition to protein–protein interactions, the mechanics of the epithelial monolayer contribute to proper localization of tricellulin to tTJs.

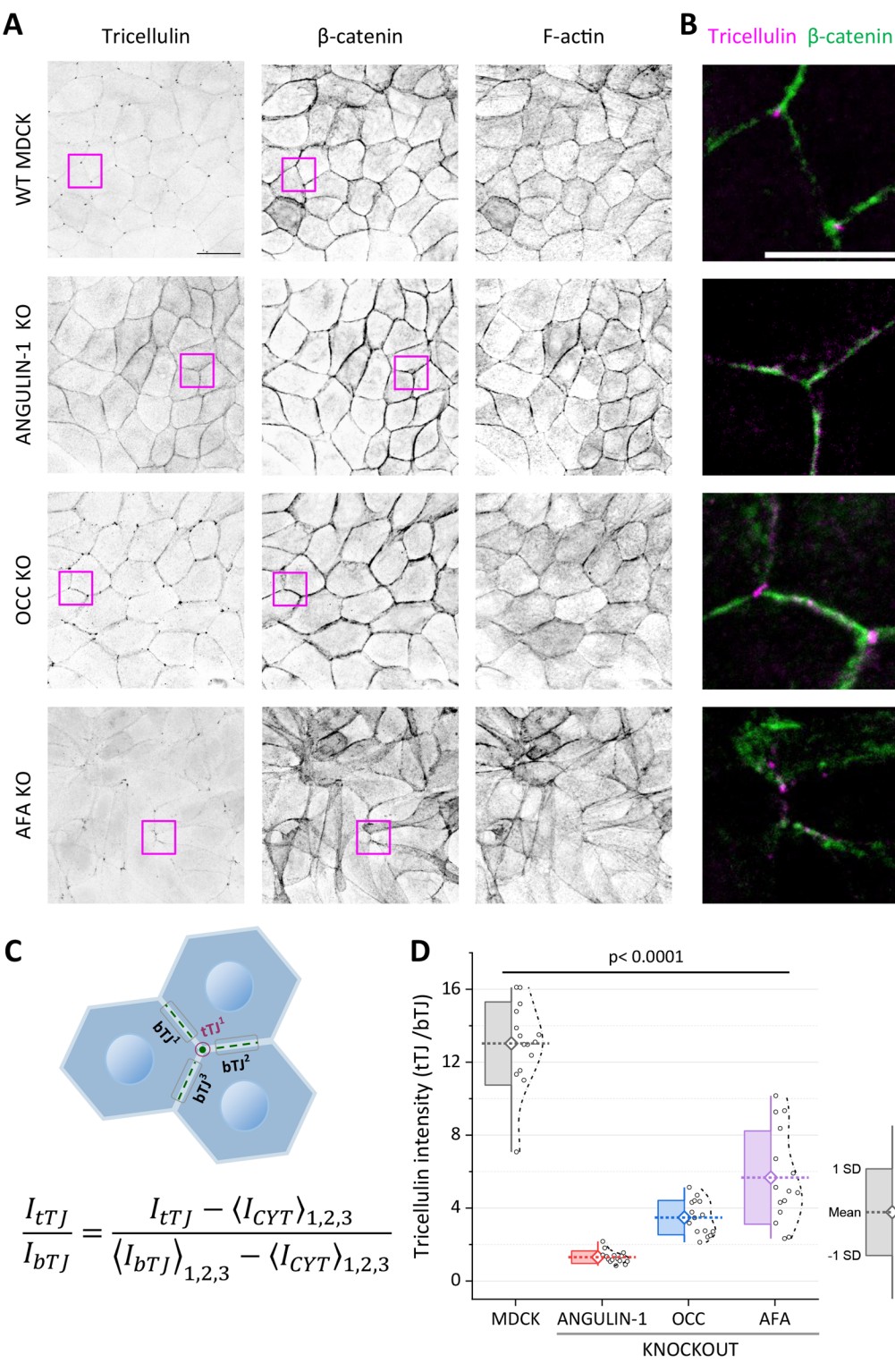

**Fig. 4. Angulin-1/LSR, occludin, and afadin knockouts displace tricellulin from tricellular junctions.** (A) Confocal microscopy images of wild-type MDCK cells, as well as angulin-1/LSR, occludin, and afadin knockout MDCK cells where tricellulin and β-catenin were detected by specific antibodies, and F-actin with phalloidin staining. Single channels are shown as inverted lookup tables (LUTs; greyscale). (B) Merged images of the boxed regions from panel A display the localization of tricellulin (magenta) in respect to cell–cell junctions, as visualized by anti-β-catenin antibody (green). Scale bars: 10 μm. (C) Schematic explaining how the enrichment ratio of tricellulin intensity at tricellular junctions ($I_{tTJs}$) compared to bicellular junctions ($I_{bTJs}$) was calculated. The intensity values were normalized to average cytoplasmic background signal ($I_{CYT}$). (D) Quantification of tricellulin enrichment in tricellular contacts versus bicellular contacts. The graph shows the enrichment ratio of tricellulin intensity at tricellular junctions (tTJs) relative to bicellular junctions (bTJs) in wild-type MDCK, as well as angulin-1/LSR, occludin, and afadin knockout cells. Data are represented as mean±s.d. Each circle and dot correspond to an individual data point for each group.

## DISCUSSION

Tricellulin localizes to tTJs, where it promotes their maintenance and epithelial barrier function (Higashi and Miller, 2017). Here, we focused on elucidating the mechanisms by which tricellulin localizes specifically to tTJs. We provide evidence that both protein–protein interactions and monolayer mechanics contribute to its accumulation to the sites where three epithelial cells meet one another.

Our fluorescence-after-photobleaching experiments revealed that tricellulin undergoes relatively rapid lateral diffusion along the bTJs

of epithelial monolayer. Its recovery half-time (t1/2∼200 s) is comparable to the one reported earlier for occludin in bTJs (t1/2∼195 s). On the other hand, another transmembrane protein, claudin, was reported to display only very slow dynamics in bTJs (Shen et al., 2008), which is consistent with claudin assembling in cell–cell contacts between adjacent cells and polymerizing to form TJ strands (Sasaki et al., 2003). Thus, we speculate that at bTJs tricellulin is not a component of a larger protein complex or protein assembly, but instead freely diffuses along the membrane plane.

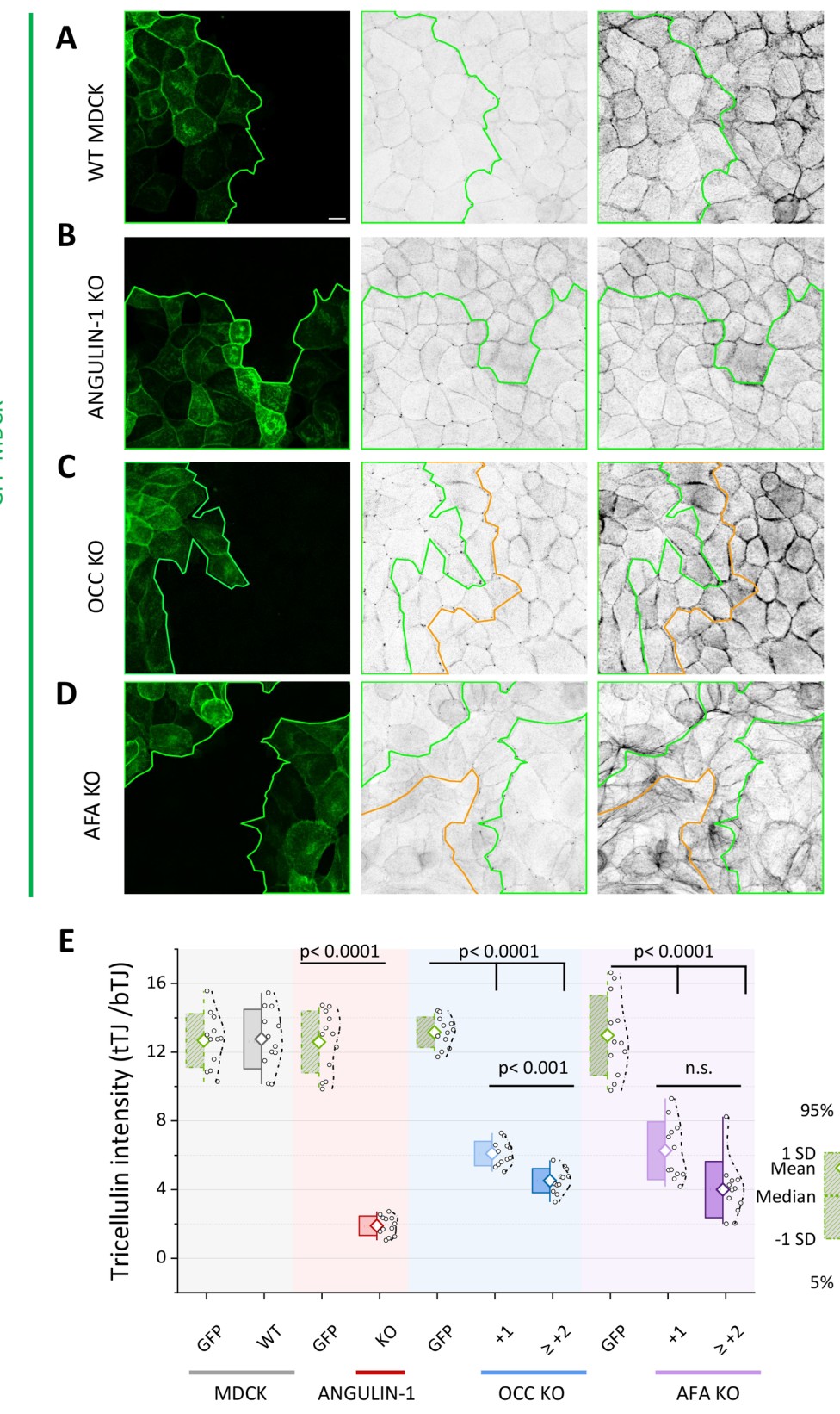

**Fig. 5. The proximity of wild-type cells influences the localization of tricellulin to tTJs in occludin knockout cells.** (A-D) Confocal microscopy images of GFP-expressing MDCK cells (green) co-cultured with 'non-fluorescent' wild-type MDCK cells (A), angulin-1/LSR knockout cells (B), occludin knockout cells (C), and afadin knockout cells (D). Tricellulin localization was detected by anti-tricellulin antibody (middle panels) and F-actin with phalloidin staining (right panels). The green lines indicate the borders between GFP-expressing and 'non-fluorescent' wild-type or knockout MDCK cell islands, and the orange lines indicate the first line of cell–cell contacts between afadin- or occludin knockout cells away from the GFP-expressing MDCK cells. Scale bars: 10 µm. (E) Quantification of enrichment of tricellulin in tTjs versus bTJs with the same method as in Fig. 4C,D. In the case of occludin and afadin knockout cells, the tricellulin ratio in tTJs versus bTJs was also quantified from the junctions between the first and second rows of knockout cells from the wild-type–knockout cell border, '+1' (orange lines in panels C and D) and in the knockout–knockout junctions further away from the GFP-expressing wild-type cells '≥+2'. Tricellulin accumulates to tTJs more efficiently in occludin knockout cells that are located in the proximity of wild-type cells. Data are represented as mean±s.d. Each circle and dot correspond to an individual data point for each group.

Importantly, our experiments revealed that tricellulin is a very stable component of tTJs, displaying hardly any recovery during the 1200 s observation period following photobleaching. This result suggests that tricellulin is a component of a very stable protein scaffold at tTJs, or alternatively the specific geometric or mechanical environment of tTJs stabilizes tricellulin specifically

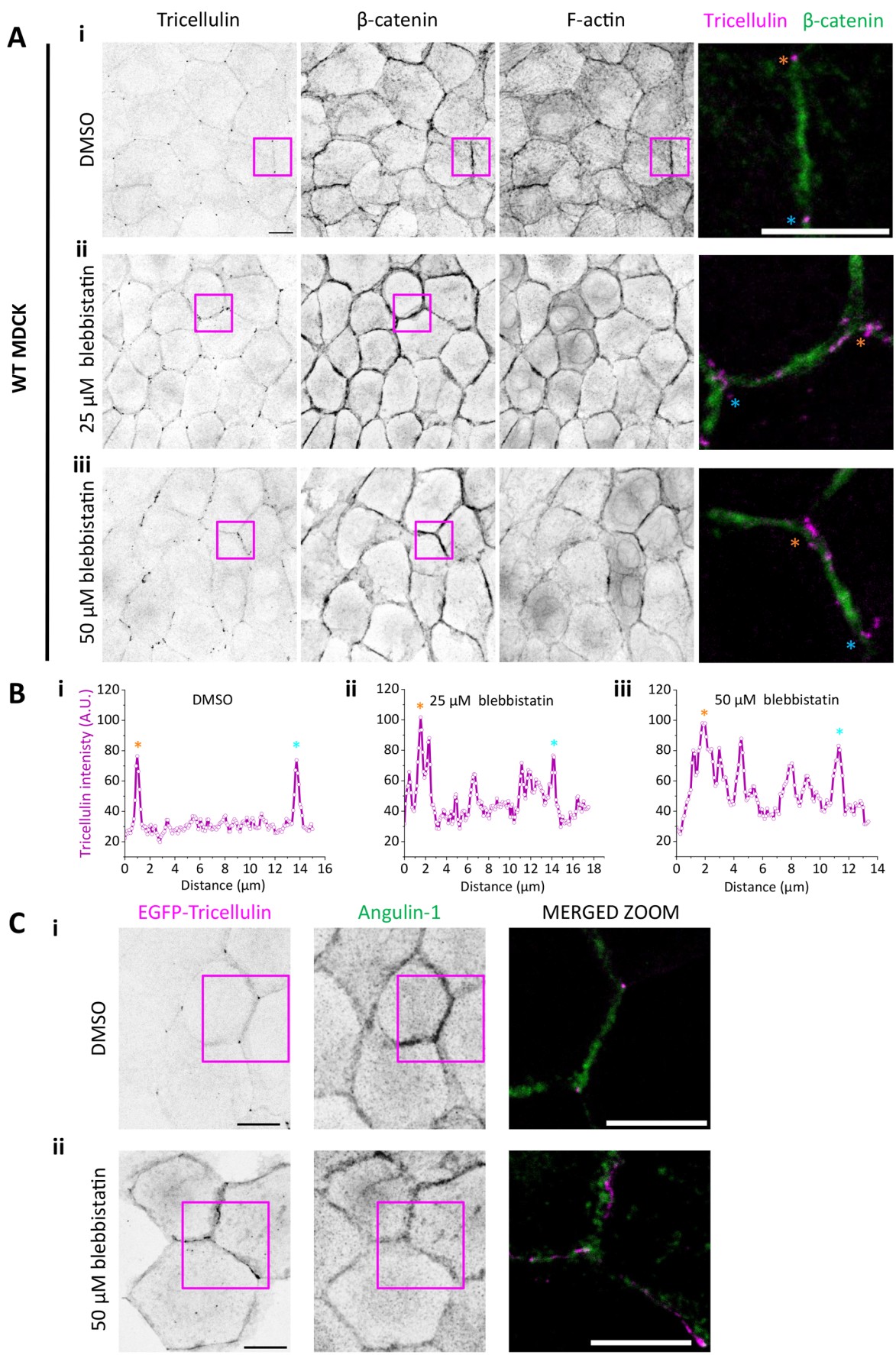

**Fig. 6.** See next page for legend.

**Fig. 6. Effects of monolayer mechanics on the localization of tricellulin.**
(A) Confocal microscopy images of wild-type MDCK monolayers treated with DMSO control (top panel), 25 µM blebbistatin (middle panel), and 50 µM blebbistatin (bottom panel), where tricellulin and β-catenin were detected by specific antibodies, and F-actin with phalloidin. Magnified merged images on right display the localization of tricellulin (magenta) in respect to cell–cell junction marker, β-catenin (green), from the boxed regions. (B) Line scans from merged images of each treatment (between the tTJs indicated with orange and cyan stars) demonstrating the effects of NM-II inhibition on the localization of tricellulin. (C) EGFP-tricellulin expressing MDCK monolayer treated with (i) DMSO and (ii) 50 µM blebbistatin, where tricellulin and angulin-1/LSR were detected with specific antibodies. Single channels are shown on the left, and merged zoomed images from the boxed regions, with tricellulin in magenta and angulin-1/LSR in green, on the right. Although EGFP-tricellulin is displaced from tTJS after blebbistatin treatment, angulin-1/LSR localization appears not drastically affected, and there is no excessive accumulation of angulin-1/LSR to tricellulin foci at bTS. Scale bars: 10 µm.

at these sites. The very slow turnover of tricellulin at tTJs revealed in our experiments on MDCK cells differs from earlier studies, which reported rather rapid fluorescence recovery of EGFP-tricellulin both in bicellular and tricellular junctions of Caco-2 cells (Raleigh et al., 2010). These differences may arise from the use of different cell lines, or from the differences between the experimental setups, including the expression levels of the fusion proteins. In our study, the FRAP experiments were carried out on cells expressing moderate levels of EGFP-tricellulin, which also displayed similar accumulation to tTJs as endogenous protein in untransfected cells.

Our BioID proteomics analysis identified several proteins that are located at close proximity to tricellulin in MDCK cell monolayers (Table 1). These include occludin, a homologue of tricellulin that localizes uniformly along the TJs. Similar to earlier studies, loss of occludin resulted in a 'leakage' of tricellulin from tTJs to bTJs (Ikenouchi et al., 2008; Kitajiri et al., 2014), but to a lesser extent than in angulin-1/LSR depleted cells. Interestingly, our co-culture experiments revealed that the proximity of wild-type cells partially rescued the 'leakage' of tricellulin to bTJs. This result suggests that occludin does not simply exclude tricellulin from bTJs as proposed in earlier publications (Ikenouchi et al., 2008; Kitajiri et al., 2014). Instead, occludin appears to control the sub-cellular localization of tricellulin through a more complex, indirect mechanism. Our BioID studies identified also the actin-binding protein afadin as a proximity-partner of tricellulin, and knockout experiments revealed that afadin is essential for proper subcellular localization of tricellulin to tTJs. It is important to note that afadin is a cytoplasmic protein that localizes uniformly along the cell–cell junctions in epithelia to control actomyosin contractility of monolayers (Sakakibara et al., 2020). Therefore, it is unlikely that afadin directly regulates the localization of tricellulin to tTJs. In line with this hypothesis, the actin cytoskeleton at cell–cell junctions of afadin knockout cells was severely disorganized, again indicating that afadin contributes to accumulation of tricellulin to tTJs through an indirect mechanism rather than via direct binding. In this context, it is also important to remember that the BioID approach does not identify direct protein–protein interactions but rather highlights those proteins that localize to the proximity of the bait. Because our BioID experiments were performed with MDCK cells grown on regular tissue culture dishes, where cells do not properly polarize, it is likely that proteins that in a native tissue environment are in different compartments are not fully separated in the experimental conditions used in the present study. Finally, our knockout studies confirmed earlier studies reporting that angulin-1/LSR is essential for proper localization of tricellulin to tTJs (Cho et al., 2022; Masuda et al., 2011; Sugawara et al., 2021). This is because specific

localization of tricellulin to tTJs was disrupted in our angulin-1/LSR knockout MDCK cell monolayers. However, our results suggest that angulin-1/LSR is not a stable interaction partner of tricellulin. This is because in MDCK monolayers grown on coverslips, angulin-1/LSR did not show similar prominent accumulation to tTJs as tricellulin (Fig. 3). This uniform accumulation of angulin-1/LSR is not due to issues with antibody stainings, because the specificity of the antibody was confirmed by wild-type/angulin-1 knockout cell co-culture experiments (Fig. S5). Moreover, angulin-1/LSR did not accumulate to tricellulin-rich foci at cell–cell junctions in blebbistatin-treated cells (Fig. 6). Finally, our BioID screens with angulin-1/LSR and tricellulin did not provide evidence of these two proteins interacting with each other (Fig. 3). Therefore, we propose that tricellulin and angulin-1/LSR associate with each other specifically only at tTJs, or that angulin-1/LSR regulates tricellulin localization to tTJs through an indirect mechanism.

Defects in tricellulin accumulation to tTJs in afadin knockout cells, and the effects of proximity of wild-type cells in tricellulin localization in occludin knockout cells suggest that also the monolayer architecture or mechanics contribute to proper localization of tricellulin to tTJs. This notion is further supported by experiments where monolayer contractility was disrupted by administration of myosin II inhibitor, blebbistatin. In the presence of blebbistatin, tricellulin typically lost its specific localization to tTJs, and instead 'leaked' to the bTJs. In this context, it is important to note that in a recent study by Cho et al., blebbistatin was reported to 'split' tricellulin from a single dot at tTJ to three short lines at bTJs, and it was suggested that loss of myosin II activity resulted in detaching the central sealing elements from one another (Cho et al., 2022). In our blebbistatin experiments we, however, often observed micron-size tricellulin 'islands' at bTJs, indicating that loss of contractility induced tricellulin leakage from tTJs to bTJs. Our FRAP experiments provided evidence that these aberrant tricellulin foci at bTJs, induced by blebbistatin treatment, were rather dynamic. This suggests that tricellulin does not assemble into stable scaffolds with other proteins in these conditions, but instead perhaps forms phase-separated entities, due, for example, to a specific lipid environment. Although the actual mechanism by which myosin II inhibition leads to displacement of tricellulin from tTJs remains to be determined, these studies demonstrate that monolayer mechanics is an important factor controlling proper subcellular localization of tricellulin. This notion is also supported by recent studies demonstrating that tricellulin associates with the actomyosin cytoskeleton via α-catenin, and that barrier function at epithelial tricellular junctions is controlled in a mechanosensitive manner via the actin-binding protein vinculin (Cho et al., 2022; van den Goor et al., 2024). We speculate that monolayer mechanics may control tricellulin localization by tension-sensitive changes in protein conformations, which may control protein–protein interactions as shown, for example, for some focal adhesion proteins (Yan et al., 2015). Alternatively, loss of contractility may affect membrane curvature at tTJs. It was reported that certain transmembrane proteins exhibit membrane curvature sensitivity and have thus a preference to accumulate to membrane regions with specific nanoscale curvature (Aimon et al., 2014). Therefore, it is also possible that myosin II inhibition may reduce the otherwise steep negative membrane curvature at tTJs to affect the localization of curvature-sensitive proteins. In this context, it is interesting to note that loss of angulin-1/LSR was reported to affect the membrane geometry at tTJs (Sugawara et al., 2021).

Together, our work provides evidence that tricellulin undergoes rapid lateral diffusion along the plasma membrane at bTJs but is

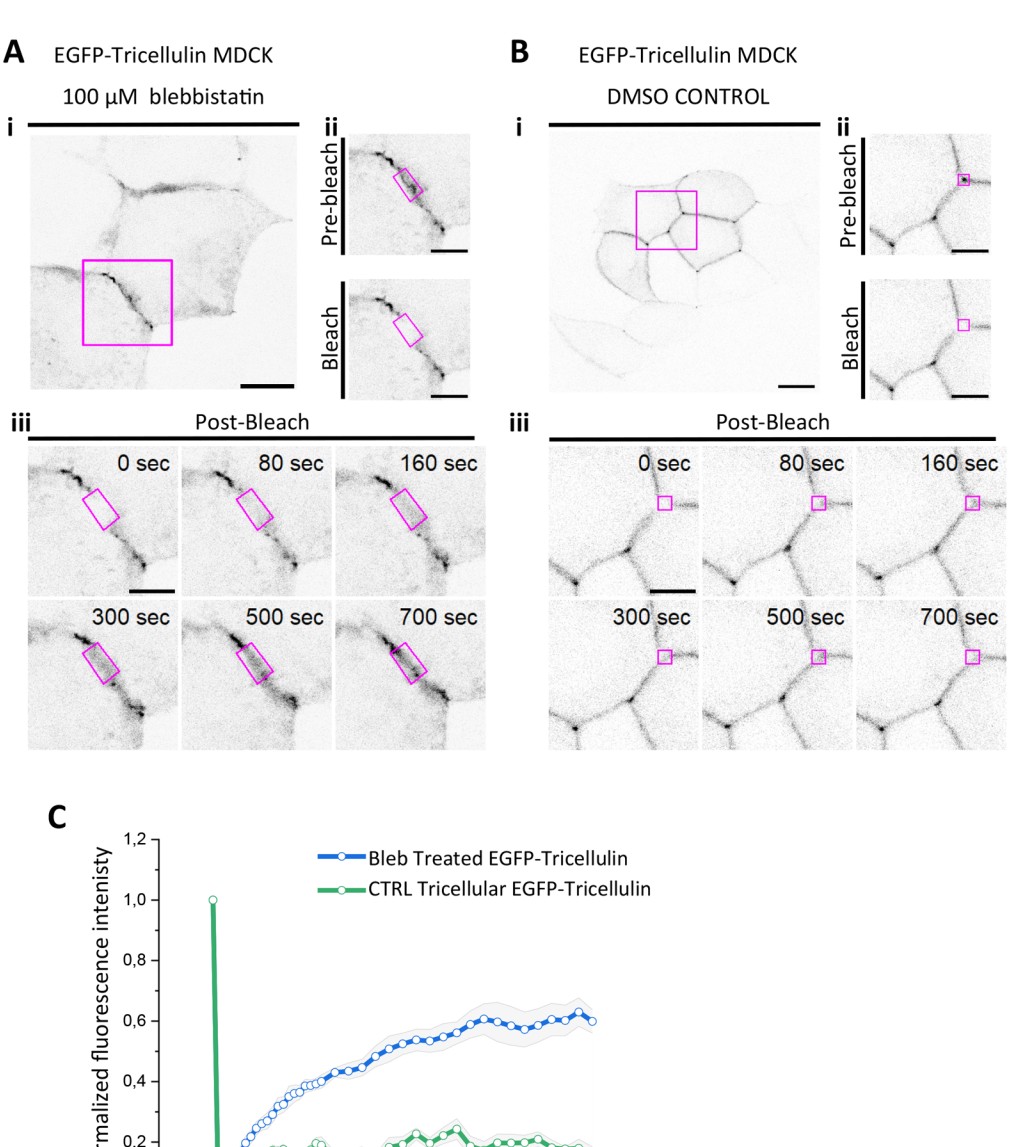

**Fig. 7. Dynamics of aberrant tricellulin foci in bicellular junctions of blebbistatin-reated monolayers.** (A,B) The dynamics of EGFP-tricellulin in confluent MDCK cell monolayers treated with 100 μM blebbistatin in DMSO (2 h, left panel) or DMSO only (control, right panel) was studied by FRAP. In blebbistatin-treated cells, the dynamics of EGFP-tricellulin were examined from aberrant foci at the bicellular junctions, whereas in the control cells the tricellulin dynamics were examined at tricellular junctions. Scale bars: 10 μm. High-magnification images of tight junction segments (Aii, Bii) display photobleached cells before and after bleaching at the indicated region (magenta boxes in whole-cell images), and the time-lapse images below (iii) show the recovery of tricellulin fluorescence in bicellular junctions of blebbistatin-treated cells (panel A) and tricellular junctions of control cells (panel B). Smaller magenta squares and rectangles represent the photobleached regions. Scale bar: 5 μm. (C) Recovery curves demonstrating that the aberrant tricellulin foci in bTJs of blebbistatin-treated cells display much more rapid dynamics than tricellulin at tTJs of control cells. The data are mean±standard error of the mean (s.e.m.) of FRAP experiments at tTJ and at bTJ (*n*=6).

trapped to tTJs in a manner that is dependent on proper monolayer mechanics or nanoscale architecture of tTJs, and the presence of angulin-1/LSR, afadin, and occludin. Our experiments also suggest that these proteins may control tricellulin localization to tTJs indirectly by regulating monolayer mechanics or geometry, instead of direct interactions with tricellulin. Further work is required to uncover the specific mechanism by which monolayer mechanics regulates the sub-cellular localization of tricellulin. It is also likely that, in addition to angulin-1/LSR, TUBA, afadin, and α-catenin, tricellulin associates with other proteins at tTJs. Our BioID screen identified several other proteins located in the proximity of tricellulin in MDCK cell monolayers. These include for example Par-3, which contributes to TJ formation together with afadin (Ooshio et al., 2007), regulators of the actin cytoskeleton, including WASH complex subunit 3 as well as ROCK1 and ROCK2, which phosphorylate myosin light chain, increasing actomyosin contractility (Totsukawa et al., 2000), and several scaffolding and

signaling proteins. Thus, future work on these proteins is likely to elucidate the structure and dynamics of tTJs, as well as their interplay with cell mechanics.

## MATERIALS AND METHODS
### Cell culture
Madin Darby canine kidney (MDCK) epithelial cells (female), and GFP-expressing stable MDCK cell lines were maintained at 37°C, 5% $CO_2$, and 95% relative humidity in minimum essential medium (MEM) supplemented with 10% fetal bovine serum (FBS; Gibco, 10500-064); 10 U/ml penicillin, 10 mg/ml streptomycin, and 20 mM L-glutamine (PSG; Gibco, 10378-016), referred to as complete media. For the BioID experiments, BioID2-HA-Tricellulin-Flp-In T-Rex, Angulin-1-HA-BioID2-Flp-In T-Rex and eGFP-BioID2-HA-CAAX-Flp-In T-Rex MDCK epithelial cells were used. These cells were maintained in MEM supplemented with Tetracycline-free FBS (ThermoFisher Scientific, SH3007003T); PSG; 100 μg/ml Hygromycin B (InvivoGen, ant-hg-1), 7 μg/ml Blasticidin S (InvivoGen, ant-bl-1). CRISPR-Cas9 knockout cell lines for tricellulin, angulin,

occludin and afadin were fluorescence activated cell (FAC)-sorted using BD FACSAria™ II Cell Sorter as single cells onto a 96-well plate, supplemented with MEM containing 10% FBS and 10 mM HEPES buffer. A construct encoding wild-type tricellulin tagged at its N-terminus with EGFP was cloned into the pEGFP-G1 vector and transfected into MDCK cells (PolyJet transfection reagent, SL100688). Stable MDCK epithelial cells expressing EFGP-tricellulin were selected and maintained in complete MEM supplemented with 450 µg/ml G418 (Merck, G418-RO). For live-cell imaging experiments, cells were cultured in MEM, no Phenol Red (ThermoFisher Scientific Gibco, 51200046) supplemented with 10% FBS and GlutaMAX (Gibco, 35050061) and 10 mM HEPES buffer. All cell lines were regularly tested for mycoplasma contamination using the Mycoalert™ Mycoplasma Detection Kit (LONZA, LT07-418). NGS was performed at the DNA Sequencing and Genomics Laboratory (BIDGEN) laboratory (Institute of Biotechnology, University of Helsinki, Finland). For treatments of cell cultures (WT MDCK and EGFP-tricellulin MDCK), myosin II-inhibitor, Blebbistatin in DMSO (Sigma-Aldrich, B0560-5MG) was used in a final concentration of 25 µM and 50 µM and incubated for 4.5 h. Inhibitor stock was made to dimethyl sulfoxide (DMSO) (Sigma-Aldrich, D4540) that was also used as control in the corresponding blebbistatin experiments.

## BioID

BioID analysis for Tricellulin and Angulin-1 was performed as described in (Kumari et al., 2020; Lehtimäki et al., 2017; Roux et al., 2012). Constructs expressing BirA fused to the N-terminus of Tricellulin (BioID2-HA-Tricellulin) and C-terminus of angulin-1 (Angulin-1-HA-BioID2) were amplified and subcloned into the pcDNA4/TO vector (ThermoFisher Scientific). An HA tag, introduced as a linker between the cDNA and BirA, was used as a complementary sequence to fuse the corresponding cDNA and BirA together in a nested PCR reaction and was further amplified in later cycles with the primers binding the 5′ and 3′ terminal ends of the fusion construct. Backbone vector expressing MDCK cells eGFP-BioID2-HA-CAAX was transfected to MDCK cells as control. For proteomics, cells displaying strong induction after tetracycline addition were expanded to five 15 cm tissue culture plates and treated with 2 µg/ml tetracycline and 50 µM biotin in complete DMEM. Cells were harvested 24 h after the induction and washed three times with PBS. Cell pellets were snap-frozen in liquid $N_2$ and stored at −80°C. Cell pellets were lysed on ice for 10 min in HNN buffer, supplemented with 0.5% NP-40, 1.5 mM Na3VO4, 1.0 mM PMSF, 10 µl/ml protease inhibitor cocktail (Sigma-Aldrich, P2714), 0.1% SDS, and 80 U/ml Benzonase Nuclease (Santa Cruz Biotechnology, Inc., Santa Cruz Biotechnology, Inc., sc-202391). Incubation was followed by three cycles of sonication. Lysate was centrifuged twice at 16,000 $g$ at 4°C to remove the insoluble material. During centrifugation, Bio-Spin chromatography columns (Bio-Rad Laboratories) were loaded with Strep-Tactin Sepharose beads (400 µl 50% slurry; IBA Lifesciences) and washed once with 1-ml HNN buffer containing 0.5% NP-40 and inhibitors (HNN wash buffer). Cleared lysate was loaded onto spin columns, followed by three washes with ice-cold HNN wash buffer with supplements and four washes with ice-cold HNN buffer without supplements. Bound proteins were eluted with 2×300-µl freshly prepared 0.5 mM d-biotin (ThermoFisher Scientific) in HNN buffer into a fresh 2-ml tube. Single-step affinity purification of the biotinylated proteins was performed as in Kumari et al., 2020 and Lehtimäki et al., 2017.

## Liquid chromatography–mass spectrometry (LC–MS)

LC-MS analysis was conducted using a Q-Exactive mass spectrometer (ThermoFisher Scientific) equipped with an EASY-nLC 1000 system and an electrospray ionization source. Peptides were separated on a C18 precolumn (Acclaim PepMap 100, 75 µm×2 cm, 3 µm, 100 Å; ThermoFisher Scientific) and an analytical column (Acclaim PepMap RSLC, 75 µm×15 cm, 2 µm, 100 Å; ThermoFisher Scientific). A 60-min gradient from 5% to 35% buffer B was followed by a 5-min gradient to 80% buffer B and a final 10-min gradient to 100% buffer B at a flow rate of 300 nl/min. Buffer A consisted of 0.1% formic acid in 98% HPLC grade water and 2% acetonitrile, while buffer B consisted of 0.1% formic acid in 98% acetonitrile and 2% water. For automated analysis, 4 µl of each peptide sample was loaded from a cooled autosampler. Data-dependent acquisition

(DDA) was performed in positive ion mode over an 80-min acquisition window. A full scan (m/z 200–2000) was conducted at a resolution of 70,000, followed by top 10 collision-induced dissociation (CID) MS/MS scans with a resolution of 17,500. Dynamic exclusion was applied for 30 s. Acquired MS/MS data files (RAW format) were processed using Proteome Discoverer 1.4 (ThermoFisher Scientific) with the SEQUEST search engine against the canine UniProtKB/SwissProt database. Search parameters included a precursor mass tolerance of ±15 ppm, fragment mass tolerance of 0.05 Da, and trypsin as the digestion enzyme, allowing up to two missed cleavages. Static carbamidomethylation of cysteine and variable modifications, including methionine oxidation and biotinylation of lysine and N-termini, were considered. High-confidence peptide identifications were filtered with a false discovery rate (FDR) of <1%.

## Antibodies and reagents

The following antibodies were used for western blotting (WB) and immunofluorescence (IF) microscopy: rabbit monoclonal anti-ZO1 (Cell Signaling Technology, 8193S, 1:100 for IF), rabbit monoclonal anti-LSR (Atlas, HPA007270, 1:150 for IF, 1:500 for WB), rabbit polyclonal anti-AFDN (Sigma-Aldrich, HPA030212, 1:50 for IF, 1:1000 for WB), rabbit monoclonal anti-MARVELD2 (ThermoFisher Scientific, 700191, 1:100 for IF, 1:1000 for WB), mouse monoclonal anti-occludin (ThermoFisher Scientific, 33-1500, 1:150 for IF, 1:1000 for WB), mouse anti-β-catenin (ThermoFisher Scientific, 13-8400, 1:100 for IF, 1:100 for WB, RRID: AB_2533039), rabbit anti-GFP (Abcam, ab290, 1:10,000 WB, RRID:AB_303421), rabbit anti-HA (Cell Signaling Technology, 3724, 1:400 for IF, RRID: AB_2256024) and goat anti-rabbit-IgG AlexaFluor-488-conjugated secondary antibodies (ThermoFisher Scientific, A11001, A-11034, 1:300 for IF, RRID:AB_2576217), goat anti-rabbit-IgG AlexaFluor-568-conjugated secondary antibodies (ThermoFisher Scientific, A-11011, 1:300 for IF, RRID: AB_143157), goat anti-rabbit-IgG AlexaFluor-647-conjugated secondary antibodies (ThermoFisher Scientific, A31571, 1:300 for IF, RRID:AB_2535813), goat anti[1]mouse-IgG AlexaFluor-568-conjugated secondary antibodies (ThermoFisher Scientific, A-11031, 1:300 for IF, RRID:AB_144696), goat anti-mouse[1]IgG HRP-conjugated secondary antibodies (ThermoFisher Scientific, 31430, 1:3000 for WB, RRID:AB_228307) and goat anti-rabbit-IgG HRP[1]conjugated secondary antibodies (ThermoFisher Scientific, 32460, 1:1000 for WB, RRID:AB_1185567). Other imaging reagents used in the study were as follows: AlexaFluor-488–phalloidin (ThermoFisher Scientific, A12379, 1:400 for IF), AlexaFluor-555–phalloidin (ThermoFisher Scientific, A34055, 1:400 for IF), AlexaFluor-568–phalloidin (ThermoFisher Scientific, A12380, 1:400 for IF) and AlexaFluor-647–phalloidin (ThermoFisher Scientific, 22287, 1:400 for IF).

## Western blotting

All cell lysates were prepared by washing the cells once with PBS and scraping them into lysis buffer (50 mM Tris-HCl pH 7.5 150 mM NaCl, 1 mM EDTA, 10% Glycerol, 1% Triton X-100) supplemented with 1 mM PMSF, 10 mM DTT, 40 µg/ml DNase I and 1 µg/ml of leupeptin, pepstatin, and aprotinin. All preparations were conducted at 4°C. Protein concentrations were determined with Bradford reagent (Bio-Rad, 500-0006) and equal amounts of the total cell lysates were mixed with Laemmli sample buffer, boiled, and run on 4%–20% gradient SDS-PAGE gels (Bio-Rad, 4561096). Proteins were transferred to nitrocellulose membrane with Trans-Blot Turbo transfer system (Bio-Rad) using Mini TGX gel transfer protocol. Membrane was blocked in either 5% milk-TBS with 0.1% Tween20 (TBS-T) or with 5% BSA for 1 h at room temperature. Primary and secondary antibodies were diluted into fresh blocking buffer overnight at 4°C and 1 h at room temperature, respectively. Proteins were detected from the membranes with Western Lightning ECL Pro substrate (PerkinElmer, NEL121001EA). Band-intensity quantification from the western blots was performed with ImageJ densitometry analysis (Schneider et al., 2012) and normalized to GAPDH protein levels.

## Generation of knockout cell lines

CRISPR knockout MDCK cell lines were generated as described previously (Senju et al., 2023). The following guide RNA sequences were used targeting MARVELD2/Tricellulin exon 1b (CGTCAACGACACCA-ACCGCG), LSR/Angulin-1 exon2 (CCCCGGCTACAACCCGTATG),

Afadin exon4 (AGGAATTCCGCAGCTCGGAT) and Occludin exon3 (TACGGGTTTGGCTACGGCTA). Transfected cells were sorted with FACSAria II (BD), as single cells onto a 96-well plate, supplemented with MEM containing 10% FBS and 10 mM HEPES buffer. For this study, two CRISPR clones were selected for each knockout based on the lack of detectable proteins by western blotting (Fig. S2A). All the data presented in the manuscript for knockouts are from one of the CRISPR clones, but the phenotypes were very similar between two clones for each knockout. The genomic DNA from wild-type and knockout MDCK cells were extracted using the Genomic DNA Extraction Kit (Invitrogen, K1820-00). The genomic region surrounding the target sequence region was amplified using primers 5′- CTCACAGAGGCCCTGCTAAC and 5′- CTGTGATGGT-GATCCTCCGG for angulin-1; 5′- GGCAGTTGGATTGACTCCCA and 5′- CCATATTTGCCTGTGTCGCCA for occludin for Sanger sequencing analysis. Moreover, Illumina Next Generation Sequencing (NGS) was performed using primers for tricellulin and afadin 5′- ACACTCTTTCCC-TACACGACGCTCTTCCGATCTGTGTTGAACAGCGGGTAGGA, 5′-GTGACTGGAGTTCAGACGTGTGCTCTTCCGATCT ACAGCGAGT-GGTACAACCTG and 5′-ACACTCTTTCCCTACACGACGCTCTTCC-GATCTCATGCCTGTAACTGTCAGCC, 5′- GTGACTGGAGTTCAGA-CGTGTGCTCTTCCGATCTATCAGATGCCTGCCTCACTG respectively. Sequence analysis was performed using Geneious version 6.1.8 (Biomatters). Both WB (Fig. S1) and sequencing data (Figs S2 and S3) confirmed complete knockout of angulin-1, occludin, tricellulin, and afadin.

### Immunofluorescence staining

MDCK cells were cultured on 10 µg/ml fibronectin-coated (Sigma-Aldrich, L2020) high precision glass coverslips. The cells were washed with 1× PBS before fixation with 4% PFA. Cell were permeabilized with 0.1% Triton X-100/TBS for 5 min, transferred to 5% BSA in PBS for 1 h at room temperature for blocking, and incubated with primary antibodies in 5% BSA for 1 h at room temperature. Cells were washed several times with PBS-T (0.02% Tween-20 in 1× PBS) and incubated with secondary antibodies and Alexa Fluor 488, 568, and 647 conjugated to phalloidin (ThermoFisher Scientific, A12379, A12380, A22287; 1:300) for 1 h at room temperature. Both primary and secondary antibodies diluted with 5% BSA were applied onto cells, incubated at room temperature in a dark, humified chamber, and washed three times with PBS-T each time post incubation. After several washes with PBS-T, coverslips were mounted onto microscopes slides with ProLong™ glass antifade mountant (ThermoFisher Scientific, P36980).

### Microscopy

#### Airyscan

For localization studies (Fig. 3), super-resolution airyscan confocal microscopy was employed. Cells were imaged using the LSM 880 (ZEISS) with a 40×1.4 numerical aperture objective. Images were acquired at 1024×1024 resolution, with 0.4-µm spacing along the z-axis. Image acquisition, initial processing and deconvolution was done using the airyscan detector and Zeiss ZEN2 software. Both fluorescent co-localization index and orthogonal projections (x-z) of tricellulin, angulin-1, and Afadin with TJ marker protein occludin were performed using ImageJ software, employing an automatic threshold method on the maximum intensity projection of semi-super resolution airyscan images. Displayed orthogonal views were also performed with ImageJ.

#### Confocal imaging

Imaging of fixed co-cultured monolayers (GFP-MDCK with non-fluorescent wild-type or knockout MDCK) was performed using the Leica TCS SP8 confocal microscope HC PL APO 63×/1.20 W motCORR CS2 objective using Leica LAS X 3.5.0, argon laser. Sample illumination with 488, 561, and 640 nm lasers detected with DD 488/561 beam splitter, PMT detectors and HC PL APO 63×/1.20 W motCORR CS2 objective. Images were acquired at 2048×2048 and 1024×1024 resolution, with 0.4-µm spacing along the z-axis and zoom factor 1,5 and 1. Maximum or sum projection images from stacks were generated and the images were analysed using Fiji/ImageJ version 1.53c. For line profile analysis of tricellulin, a line width of 10px was used, and lines were drawn along cell–cell junctions. The fluorescence intensity along the drawn line was measured and plotted as a

function of distance. Peak intensities indicate the concentration of tricellulin at tTJs, as well as its mis-localization along bTJ.

### Fluorescence-recovery-after-photo bleaching (FRAP)

Cells were cultured on 10 µg/ml fibronectin-coated, glass-bottomed, 35-mm dishes (Greiner Bio-One) in complete MEM supplemented with 10 mM Hepes. Before imaging the media was exchanged to MEM, no Phenol Red (ThermoFisher Scientific Gibco, 51200046) supplemented with 10% FBS and GlutaMAX (Gibco, 35050061) and 10 mM HEPES buffer. Time-lapse image series of EGFP-tricellulin MDCK were acquired using the Leica TCS SP8, equipped with Leica LAS X 3.5.0, LIS (Life Imaging Services) incubation system (The Cube, The Box, and The Brick) set at +37°C and 5% CO2, argon laser with DD 488/561 beam splitter, PMT detectors and HC PL APO 63×/1.20 W motCORR CS2 objective and FRAP booster. Acquisition was performed using Leica LAS AF version 2.8.8 and image analysis was performed using either Leica LAS AF version 2.8.8 or Fiji/ImageJ version 1.53c as described (Kokate et al., 2022). All FRAP data were imaged as follows: three frames were imaged using low laser power (488/35 mV, 1–5% laser power) and EGFP-tricellulin at tTJ and bTJs were then bleached with a single 3 ms pulse of A488 laser at full laser power. Acquisition was performed using AcquireSR (Cytiva). An A488 laser was used at 35% laser power with a 50 ms exposure time. Imaging was performed every 25 s time-lapse intervals for the total duration of 20 min. FRAP experiments on blebbistatin-treated EGFP-tricellulin MDCK cells were performed using the same method, but the cells were subjected to 100 µM blebbistatin for 2 h at +37°C. Inhibitor stock was made to dimethyl sulfoxide (DMSO) (Sigma-Aldrich, D4540), which was also used as control in the corresponding blebbistatin experiments.

### Data quantification

For FRAP experiments on EGFP-tricellulin MDCK cells (Figs 1E and 7C), data were analyzed as follows: first, background subtraction was performed by dividing the fluorescence within the region of interest at cell–cell junction by that of an identical-sized region in the cytoplasmic region of the cell. To measure the fluorescence recovery, the intensity of the bleached area was normalized to adjacent non-bleached region at tTJ or bTJ. Normalized values obtained from experimental repeats were applied to calculate the mean recovery and variation across replicates. Data were analyzed using Microsoft Excel, and fluorescence recovery curves were plotted in OriginPro (version 2018), OriginLab Corporation. https://www.originlab. com/. To quantify the enrichment of tricellulin at tTJs relative to bTJs, fluorescence intensities were measured and normalized to the cytoplasmic background of the same cell (Fig. 4C). The intensity at the tricellulin at tTJ ($I_{tTJs}$) and bTJ ($I_{bTJs}$) was normalized by subtracting the average cytoplasmic intensity ($I_{CYT}$) of the same cell. The normalized tTJ intensity was divided by the average normalized intensity from three bTJs of the same cell, providing a ratio that reflects tricellulin distribution between tTJ and bTJs ($I_{tTJs}/I_{bTJs}$). The graph includes mean, median, standard deviations, and the 95% confidence intervals for each group, illustrating the variability and statistical reliability of the data.

### Statistics

All statistical data analyses for bar graphs and FRAP data were performed with Excel (Microsoft) and Origin (Origin 2018b). The statistical analysis and graph construction for line profile analysis, a line width of 10px was used, and lines were drawn along bicellular junctions. All remaining box plots were constructed with OriginPro 2022b (OriginLab Corp.). Normality of the data were examined with the Shapiro–Wilk test and a quantile–quantile plot. For the pair of datasets following normal distribution, Student's two-sample unpaired *t*-test was used. If data did not follow normal distribution, Mann–Whitney *U*-test for two independent samples was conducted. The differences between the wild-type and GFP-expressing MDCK cells were compared using one-way ANOVA, and found to be statistically similar. The significance analysis of INTeractome (SAINT)-express tool (version 3.6) was used to statistically identify high-confidence protein–protein interactions from the BioID mass spectrometry data. Eight BioID targged GFP control runs were used as controls for comparative analysis. Proteins were retained as high-confidence interactors (HCIs) if they achieved a SAINT score of ≥0.73, corresponding to a Bayesian FDR about 0.05.

## Acknowledgments
We thank the Institute of Biotechnology Light Microscopy Unit (LMU) for technical advice and support with microscopy, and Mirva Tirkkonen for excellent technical assistance. Jaana Träskelin is acknowledged for expert technical assistance at the Biocenter Oulu Virus Core Laboratory.

## Competing interests
The authors declare no competing or financial interests.

## Author contributions
Conceptualization: P.L., T.M., J.L.; Data curation: K.K.; Formal analysis: T.M.; Funding acquisition: P.L.; Investigation: T.M., J.L., X.L., A.M.; Methodology: T.M., J.L., K.K., X.L., A.M.; Supervision: P.L., J.L., J.P., M.V.; Writing – original draft: P.L., T.M.; Writing – review & editing: J.L., K.K., J.P., X.L., M.V., A.M.

## Funding
This study was supported by the Research Council of Finland BarrierForce center of excellence funding (346133 to P.L.) and by grants from the Sigrid Juselius foundation (4708344 to P.L.) and the Cancer Society Finland (to P.L.). T.M. was supported by a fellowship from the University of Helsinki Doctoral Program of Integrative Life Sciences. Open Access funding provided by the Research Council of Finland. Deposited in PMC for immediate release.

## Data and resource availability
The reagents resulting from this study and the raw data are available from the corresponding author through a reasonable request. All other relevant data and details of resources can be found within the article and its supplementary information.

## Peer review history
The peer review history is available online at https://journals.biologists.com/bio/article-lookup/doi/10.1242/bio.061987.reviewer-comments.pdf.

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
