## [Peer Review File · Biology Open]

Roles of protein-protein interactions and monolayer mechanics in tricellulin localization to tricellular tight junctions

Toiba Mushtaq, Jaakko Lehtimäki, Konstantin Kogan, Johan Peränen, Xiaonan Liu, Markku Varjosalo, Aki Manninen and Pekka Lappalainen
DOI: 10.1242/bio.061987

Editor: Cathy Jackson

Review timeline

Original submission:	3 February 2025
Editorial decision:	13 March 2025
First revision received:	12 August 2025
Accepted:	14 August 2025

Original submission

First decision letter

MS TITLE: Roles of protein-protein interactions and monolayer mechanics in tricellulin localization to tricellular tight junctions

AUTHORS: Pekka Lappalainen; Toiba Mushtaq; Jaakko Lehtimäki; Konstantin Kogan; Johan Peränen; Xiaonan Liu; Markku Varjosalo; Aki Manninen

Dear Dr Lappalainen,

We have now reached a decision on the above manuscript.

To see the reviewers' reports and a copy of this decision letter, please go to:

Your manuscript was considered by three expert reviewers and, unfortunately, there was strong consensus among the reviewers that there was an insufficient conceptual advance. I am very sorry to give you such disappointing news, but we are currently under great pressure for space and it takes a very enthusiastic recommendation by the referees for a manuscript to be accepted. I do hope you find the comments of the reviewers helpful in allowing you to revise the manuscript for submission elsewhere, and many thanks for sending your work to us.

Reviewer 1: SUMMARY OF THE ADVANCE MADE IN THIS PAPER AND ITS POTENTIAL SIGNIFICANCE TO THE FIELD

This manuscript by Mushtaq T et al. investigates tricellular tight junctions, which are crucial for maintaining epithelial barrier function, integrity, and tissue remodeling. The authors performed FRAP analysis of tricellulin, demonstrating that it becomes highly stable only when localized at tricellular contact sites. Using a BioID method, they explored putative interacting molecules near tricellulin and angulin-1. Among these, they analyzed occludin, angulin-1, and afadin, showing that each molecule influences tricellulin localization. Finally, they examined the effect of a myosin II inhibitor on tricellulin localization.

SUGGESTIONS TO AUTHORS

Overall, the study is well-conducted, with properly controlled experiments and high-quality data. The manuscript is also well-written. However, the novelty of the findings is limited, and the contribution to the research field is not significant enough to justify publication in this journal.

Major Points

1. The FRAP analysis of tricellulin has already been performed by Raleigh et al. (MBoC, 2010). Although the results obtained in this study differ from those reported by Raleigh et al., the authors should provide a more thorough discussion comparing and contrasting these differences.
2. The BioID experiment used an angulin-1 construct with BirA fused to its N-terminus. However, angulin-1 is a type I membrane protein with a signal sequence following the start codon. Since signal sequences are typically cleaved, it is unlikely that BirA is properly fused to the mature form of angulin-1. Furthermore, placing BirA in the extracellular domain raises concerns about whether activated biotin can reach intracellular proteins, making it unclear whether intracellular interacting molecules were correctly identified.
3. The loss or reduction of tricellulin localization at tricellular contacts due to the deletion of angulin-1 or occludin has already been reported multiple times. While the observation that the knockout (KO) phenotype is slightly attenuated at the boundary with wild-type cells is novel, the current classification into "border" and "further away" is too simplistic. A more detailed analysis, such as distinguishing the following tricellular junction compositions, could provide mechanistic insights:

Within wild-type cells: WT/WT/WT, WT/WT/KO, WT/KO/KO

Within KO cells: WT/WT/KO, WT/KO/KO, KO/KO/KO

4. The experiments using a myosin II inhibitor are rather superficial. Additional experiments, such as performing FRAP under inhibitor treatment, could help elucidate the underlying molecular mechanisms. Moreover, since blebbistatin loses its inhibitory activity upon exposure to blue light (Sakamoto et al., Biochemistry, 2005), the authors could utilize localized inhibition release during live imaging to achieve more detailed analysis.

Minor Points

1. In the second paragraph, the authors state that MDCK II cells were used by Ikenouchi et al. However, Ikenouchi et al. actually used EpH4 cells for the knockdown experiment. MDCK II cells were used by Van Itallie et al. (J Cell Sci, 2010), who reported no barrier function abnormalities. Raleigh et al. (MBoC, 2010) observed some effects.
2. In the first line of page 4, the manuscript discusses tricellulin mutations causing DFNB49-related hearing loss but does not cite the original study by Riazuddin et al. (Am J Hum Genet, 2006). This citation should be included.
3. The expression of tricellulin-GFP does not appear uniform; some cells show variation in intensity. It would be preferable to use a more homogeneous population. Additionally, the authors should compare the expression levels and localization of tricellulin-GFP with endogenous tricellulin using immunostaining or Western blot analysis. However, the rabbit monoclonal anti-tricellulin antibody used in this study (clone 54H19L38) recognizes a region near the C-terminus, meaning C-terminal tagging may prevent detection.
4. A lookup table should be included for Fig. 1D to enhance clarity.
5. FRAP experiments should be repeated multiple times, and recovery half-time and mobile fraction should be presented as mean \pm SD.

6. The localization and expression levels of the molecules used in the BioID experiments should be demonstrated, for example, by using HA-tag labeling.
7. On page 6, line 2, the term "tTJs" is used. However, tricellular tight junctions (tTJs) do not exist in *Drosophila*. This should be corrected.
8. On page 13, line 10 from the bottom, "NO2" should be corrected to "N2".
9. The proteomic analysis using LC-MS employs a human database, but MDCK cells are derived from a dog. Since canine and human peptide sequences often differ, a canine protein database should be used for more accurate identification.

Reviewer 2: SUMMARY OF THE ADVANCE MADE IN THIS PAPER AND ITS POTENTIAL SIGNIFICANCE TO THE FIELD

A deeper understanding of the molecular mechanisms which regulate the localization of tricellulin at tricellular TJs is clearly significant in the field of TJ biology. One problem with the study is that the BioID experiment was performed with MDCK cells grown on regular tissue culture dishes. Obviously, this is necessary for proteomics experiments. Unfortunately, cells do not polarize properly under these conditions. It is, thus, not surprising that proteins which are in separate compartments in fully polarized cells (TJs vs AJs, tTJs vs bTJs) are in close proximity when cultured on regular culture dishes. The study appears, therefore, somewhat premature. While it is important to identify proteins that interact with tricellulin as a basis for its specific localization at tTJs, the identified binding partners do not add information on the possible mechanism. The data rather suggest that the tTJ-specific localization of tricellulin is regulated by actomyosin-mediated tension (as indicated by the blebbistatin experiment), not by the putative interaction with any of the identified binding partners. In summary, the advancement in our understanding of tricellulin localization is rather limited.

SUGGESTIONS TO AUTHORS

A deeper understanding of the molecular mechanisms which regulate the localization of tricellulin at tricellular TJs is clearly significant in the field of TJ biology. One problem with the study is that the BioID experiment was performed with MDCK cells grown on regular tissue culture dishes. Obviously, this is necessary for proteomics experiments. Unfortunately, cells do not polarize properly under these conditions. It is, thus, not surprising that proteins which are in separate compartments in fully polarized cells (TJs vs AJs, tTJs vs bTJs) are in close proximity when cultured on regular culture dishes. The study appears, therefore, somewhat premature. While it is important to identify proteins that interact with tricellulin as a basis for its specific localization at tTJs, the identified binding partners do not add information on the possible mechanism. The data rather suggest that the tTJ-specific localization of tricellulin is regulated by actomyosin-mediated tension (as indicated by the blebbistatin experiment), not by the putative interaction with any of the identified binding partners. In summary, the advancement in our understanding of tricellulin localization is rather limited.

Reviewer 3: SUMMARY OF THE ADVANCE MADE IN THIS PAPER AND ITS POTENTIAL SIGNIFICANCE TO THE FIELD

In this manuscript, Mushtaq et al. attempt to elucidate the mechanism of tricellulin localization to tricellular junctions in MDCK cells. The authors reproduce the roles of angulin-1 and occludin in tricellulin localization, and show that tricellular junction localization of tricellulin is impaired in afadin knockout cells with abnormal F-actin organization, concluding that epithelial monolayer mechanics is required for tricellulin localization. The authors also show that tricellulin molecules incorporated in to tricellular junctions are considerable stable in a FRAP assay. Overall, the immunolocalization experiments are good and the results are very carefully interpreted. However, the authors' conclusion is still preliminary to understand the mechanism of tricellulin localization although mechanics is a very interesting point to consider the structure and function of tricellular junctions. Furthermore, the study feels narrow in scope in its current form. It is largely a

localization analysis of tricellulin, without functional data that would make the story of broad interest to the general cell biology audience of JCS, such as the ultrastructure, dynamic behavior or barrier function of tricellular tight junctions.

SUGGESTIONS TO AUTHORS

1. Although the authors reproduced that angulin-1 is essential for tricellulin localization at tricellular junctions, they could not identify angulin-1 as a possible intracellular interacter of tricellulin in the BioID-screen. This may be explained by the experimental condition, in which biotin ligase was fused to the N-terminus of tricellulin. A previous study showed that the C-terminal cytoplasmic domain of tricellulin interacts with the cytoplasmic region of angulin-1 (Masuda et al. 2011). This observation needs to be mentioned. As far as this reviewer knows, it has not been clarified whether the interaction between angulin-1 and tricellulin is direct or indirect.

2. Page 6, line 5. Did the authors really perform BioID screen with N-terminally tagged angulin-1? The N-terminus of angulin-1 is exposed to the extracellular space and cannot interact with actomyosin-related components.

3. The authors state that depletion of occludin resulted in diminished localization of tricellulin to tTJs and show the tTJ/bTJ of tricellulin intensity in Figs 4 and 5. Does the tricellulin signal at tricellular junction decrease in occludin KO cells and afadin KO cells? Another possibility is that the intensity of tricellulin at tricellular junctions does not change, but diffusely distributed tricellulin throughout the lateral membrane become concentrated into bicellular tight junctions in these cells.

Minor

1. Page 3, line 5 from the bottom. This study (Ikenouchi et al., 2005) used Eph4 cells instead of MDCK II cells. Here, the author should refer to another study by Sugawara et al. (2021), in which tricellulin knockout influence neither the epithelial barrier function nor the main structure of tricellular tight junctions in MDCK II cells.

2. About immunostaining of angulin-1, is there a possibility that the tricellular junction associated angulin-1 signal cannot be clearly identified because it is surrounded by considerably strong signals of angulin-1 along the lateral membrane?

First revision

Author response to reviewers' comments

Reviewer #1:

SUGGESTIONS TO AUTHORS

Overall, the study is well-conducted, with properly controlled experiments and high-quality data. The manuscript is also well-written. However, the novelty of the findings is limited, and the contribution to the research field is not significant enough to justify publication in this journal.

Major Points

1. The FRAP analysis of tricellulin has already been performed by Raleigh et al. (MBoC, 2010). Although the results obtained in this study differ from those reported by Raleigh et al., the authors should provide a more thorough discussion comparing and contrasting these differences.

We have now included more discussion on page 10 to compare the differences between the FRAP data of our study and the work by Raleigh et al., (2010).

2. The BioID experiment used an angulin-1 construct with BirA fused to its N-terminus. However, angulin-1 is a type I membrane protein with a signal sequence following the start codon. Since signal sequences are typically cleaved, it is unlikely that BirA is properly fused to the mature form of angulin-1. Furthermore, placing BirA in the extracellular domain raises concerns about whether activated biotin can reach intracellular proteins, making it unclear whether intracellular interacting molecules were correctly identified.

Thank you for pointing out this error in the manuscript text! Angulin-1 was tagged to its C-terminus (not to the N-terminus as mistakenly stated in the original version of our manuscript), and this is now corrected throughout the manuscript text. Additionally, we present new data (Fig. 2 A-D) demonstrating the expression levels and subcellular localizations of the tricellulin and angulin-1 BirA-fusion constructs.

We also identified few other errors regarding the description of constructs used in this study, and these are now corrected in the revised manuscript.

3. The loss or reduction of tricellulin localization at tricellular contacts due to the deletion of angulin-1 or occludin has already been reported multiple times. While the observation that the knockout (KO) phenotype is slightly attenuated at the boundary with wild-type cells is novel, the current classification into "border" and "further away" is too simplistic. A more detailed analysis, such as distinguishing the following tricellular junction compositions, could provide mechanistic insights:

Within wild-type cells: WT/WT/WT, WT/WT/KO, WT/KO/KO
 Within KO cells: WT/WT/KO, WT/KO/KO, KO/KO/KO

We apologize that the classification (border/further away) was not particularly clearly explained in the original version of the manuscript. We have now modified Fig. 5 and its legend to better explain the definitions of different cell-cell junctions analyzed here. For clarity, we also replaced "BRD" by "+1" in the figure, and explain this in more detail in the figure legend. Furthermore, we replaced "FAR" in the figure by "≥+2" and explain this in more detail in the figure legend.

As requested, we also analyzed junctions between occludin knockout cells at different distances away from the wild-type/knockout border. However, because the effects of wild-type cells did not clearly propagate to these 'more distant' junctions, and because the main result (the proximity of wild-type cells affects the subcellular localization of tricellulin in the occludin knockout cells) is already evident from the analysis of "+1" junctions, we did not include these data to the manuscript. If considered necessary, we can prepare an additional supplementary figure from these data.

4. The experiments using a myosin II inhibitor are rather superficial. Additional experiments, such as performing FRAP under inhibitor treatment, could help elucidate the underlying molecular mechanisms. Moreover, since blebbistatin loses its inhibitory activity upon exposure to blue light (Sakamoto et al., Biochemistry, 2005), the authors could utilize localized inhibition release during live imaging to achieve more detailed analysis.

This is a very good suggestion, and we have accordingly carried out FRAP experiments on blebbistatin-treated cells. These experiments revealed that the tricellulin displays rapid lateral diffusion in the blebbistatin-induced aberrant foci at bi-cellular junctions. These new data (shown in new Fig. 7 and discussed on page 9) provide further evidence for the role of monolayer mechanics in regulating lateral diffusion/dynamics of tricellulin at tricellular cell-cell junctions.

Minor Points

1. In the second paragraph, the authors state that MDCK II cells were used by Ikenouchi et al. However, Ikenouchi et al. actually used EpH4 cells for the knockdown experiment. MDCK II cells were used by Van Itallie et al. (J Cell Sci, 2010), who reported no barrier function abnormalities. Raleigh et al. (MBoC, 2010) observed some effects.

Thank you for pointing this out. We have now corrected this and included the Van Itallie et al., (2010) reference to the text. Moreover, as suggested by reviewer #3, we also included Sugawara et al., (2021) reference here.

2. In the first line of page 4, the manuscript discusses tricellulin mutations causing DFNB49-related hearing loss but does not cite the original study by Riazuddin et al. (Am J Hum Genet, 2006). This citation should be included.

This is now included in the manuscript text.

3. The expression of tricellulin-GFP does not appear uniform; some cells show variation in intensity. It would be preferable to use a more homogeneous population. Additionally, the authors should compare the expression levels and localization of tricellulin-GFP with endogenous tricellulin using immunostaining or Western blot analysis. However, the rabbit monoclonal anti-tricellulin antibody used in this study (clone 54H19L38) recognizes a region near the C-terminus, meaning C-terminal tagging may prevent detection.

The expression levels of EGFP-tricellulin vary between individual cells in our stable cell lines. However, we have now performed Western blot analysis on the EGFP-tricellulin cell line. These new data, shown in Fig. S1A-B, suggest that the expression level of EGFP-tricellulin in this cell-line is comparable to the levels of endogenous tricellulin in the same cell population.

4. A lookup table should be included for Fig. 1D to enhance clarity.

This is now included in the figure.

5. FRAP experiments should be repeated multiple times, and recovery half-time and mobile fraction should be presented as mean \pm SD.

We have now repeated the FRAP experiments in Fig. 1 for several times (n=8) both for tricellular and bicellular junctions, and accordingly prepared a new graph (Fig. 1D in the revised manuscript). We also agree that in many cases it is informative to include a graph displaying half-times and mobile fractions of the FRAP data. However, because tricellulin displays only 10-20 % fluorescence recovery at tricellular junctions during the 20 min imaging period, it is not possible to present a recovery half-life (or mobile fraction) in this case. Moreover, presenting mobile fractions of FRAP experiments is often not particularly informative, because nearly all proteins display some mobility in cellular structures, but may display different populations with distinct dynamic properties (i.e. there are no specific mobile and immobile fractions, but just fractions with different mobilities).

6. The localization and expression levels of the molecules used in the BiID experiments should be demonstrated, for example, by using HA-tag labeling.

This is a good point, and we have now examined the sub-cellular localizations of tricellulin and angulin-1 BirA-constructs. BirA-Tricellulin localizes predominantly to cell-cell junctions with enrichment in tricellular junctions. Also angulin-1-BirA localized to cell-cell junctions, although a fraction of the protein was also found in cytoplasmic puncta (which may represent e.g. ER). These new data are shown in Fig. 2 A-D.

7. On page 6, the term "tTJs" is used. However, tricellular tight junctions (tTJs) do not exist in Drosophila. This should be corrected.

Thank you! This is now corrected.

8. On page 14, "NO2" should be corrected to "N2".

This is corrected in the revised manuscript.

9. The proteomic analysis using LC-MS employs a human database, but MDCK cells are derived from a dog. Since canine and human peptide sequences often differ, a canine protein database

should be used for more accurate identification.

We apologize for this mistake. In the analysis, we used a canine database and this is now corrected in the 'Methods'. We have now also included the BioID analysis data file as a supplementary Excel table.

Reviewer 2:

SUGGESTIONS TO AUTHORS

A deeper understanding of the molecular mechanisms which regulate the localization of tricellulin at tricellular TJs is clearly significant in the field of TJ biology. One problem with the study is that the BioID experiment was performed with MDCK cells grown on regular tissue culture dishes. Obviously, this is necessary for proteomics experiments. Unfortunately, cells do not polarize properly under these conditions. It is, thus, not surprising that proteins which are in separate compartments in fully polarized cells (TJs vs AJs, tTJs vs bTJs) are in close proximity when cultured on regular culture dishes.

We agree that ideally one should carry out BioID experiments in a more physiological tissue environment, but for technical reasons we chose to do our angulin-1 and tricellulin BioID experiments on cell monolayers grown on tissue culture dishes. We have now added discussion (page 11) to explain the disbenefits of this approach (including the fact that different compartments are not well separated in our system).

The study appears, therefore, somewhat premature. While it is important to identify proteins that interact with tricellulin as a basis for its specific localization at tTJs, the identified binding partners do not add information on the possible mechanism. The data rather suggest that the tTJ-specific localization of tricellulin is regulated by actomyosin-mediated tension (as indicated by the blebbistatin experiment), not by the putative interaction with any of the identified binding partners. In summary, the advancement in our understanding of tricellulin localization is rather limited.

The most important finding of our study is, indeed, that actomyosin contractility is critical for proper localization of tricellulin to tricellular junctions. To further address this point, we have now added more discussion about the contribution of actomyosin contractility and possible roles of protein-protein interactions on the subcellular localization of tricellulin on page 12. Furthermore, we have carried out additional FRAP experiments (Fig. 7) to further examine the effects of monolayer mechanics on tricellulin dynamics.

Reviewer 3:

SUGGESTIONS TO AUTHORS

1. Although the authors reproduced that angulin-1 is essential for tricellulin localization at tricellular junctions, they could not identify angulin-1 as a possible intracellular partner of tricellulin in the BioID-screen. This may be explained by the experimental condition, in which biotin ligase was fused to the N-terminus of tricellulin. A previous study showed that the C-terminal cytoplasmic domain of tricellulin interacts with the cytoplasmic region of angulin-1 (Masuda et al. 2011). This observation needs to be mentioned. As far as this reviewer knows, it has not been clarified whether the interaction between angulin-1 and tricellulin is direct or indirect.

These are good points, and we have now added discussion about tagging tricellulin to N- vs. C-terminus, as well as on direct/indirect interaction between tricellulin and angulin-1 (see page 6, and page 12).

2. Page 6, line 5. Did the authors really perform BioID screen with N-terminally tagged angulin-1? The N-terminus of angulin-1 is exposed to the extracellular space and cannot interact with

actomyosin-related components. line18

We apologize for an error in describing the angulin-1 BioID construct. In this construct the BirA was tagged to the C-terminus of angulin-1, and this is now corrected throughout the text. We have now also included new data, which show the expression levels and sub-cellular localizations of the tricellulin and angulin-1 BirA constructs (Fig. 2 A-D).

We also identified some other errors in description of the constructs used in this work, and these are corrected in the revised manuscript.

3. The authors state that depletion of occludin resulted in diminished localization of tricellulin to tTJs and show the tTJ/bTJ of tricellulin intensity in Figs 4 and 5. Does the tricellulin signal at tricellular junction decrease in occludin KO cells and afadin KO cells? Another possibility is that the intensity of tricellulin at tricellular junctions does not change, but diffusely distributed tricellulin throughout the lateral membrane become concentrated into bicellular tight junctions in these cells.

We have now compared the intensities of tricellulin in tricellular junctions of wild-type, occludin KO and afadin KO cells. This analysis provided very similar results to our original data where we compared the distribution of tricellulin between tricellular and bicellular junctions (i.e. the tricellulin intensity at tricellular junctions decreases in the knockout cells). These analysis are shown in the figure below. Because these data are somewhat repetitive with each other, we chose not to include the new quantifications to the manuscript. Nevertheless, if considered necessary, we can include these new data as an additional supplementary figure.

Minor

1. Page 3, line 5 from the bottom. This study (Ikenouchi et al., 2005) used Eph4 cells instead of MDCK II cells. Here, the author should refer to another study by Sugawara et al. (2021), in which tricellulin knockout influence neither the epithelial barrier function nor the main structure of tricellular tight junctions in MDCK II cells.

Thank you for pointing out this error, which is now corrected.

2. About immunostaining of angulin-1, is there a possibility that the tricellular junction associated angulin-1 signal cannot be clearly identified because it is surrounded by considerably strong signals of angulin-1 along the lateral membrane?

Our data indeed show that there is a considerably strong signal of angulin-1 at the bicellular junction (lateral membrane). Thus, angulin-1 is also present at tricellular junctions, but, unlike tricellulin, it is not concentrated there. This is further clarified on page 6.

Second decision letter

MS ID#: bio.061987

MS Title: Roles of protein-protein interactions and monolayer mechanics in tricellulin localization to tricellular tight junctions

Authors: Pekka Lappalainen; Toiba Mushtaq; Jaakko Lehtimäki; Konstantin Kogan; Johan Peränen; Xiaonan Liu; Markku Varjosalo; Aki Manninen

Dear Dr Lappalainen,

I am happy to tell you that your manuscript has been accepted for publication in Biology Open, pending our standard publication integrity checks. It was accepted on 14th August 2025. The

responses to all of the reviewers' comments and addition of new experiments now provide data that support the interesting conclusions made.